# The effects of type and workload of internal tasks on voluntary saccades in a target-distractor saccade task

**Sonja Walcher**[1]*, **Živa Korda**[1], **Christof Körner**[2], **Mathias Benedek**[1]

1 Creative Cognition Lab, Institute of Psychology, University of Graz, Graz, Austria, 2 Cognitive Psychology & Neuroscience, Institute of Psychology, University of Graz, Graz, Austria

* sonja.walcher@uni-graz.at

## Abstract

When we engage in internally directed cognition, like doing mental arithmetic or mind wandering, fewer cognitive resources are assigned for other activities like reacting to perceptual input—an effect termed perceptual decoupling. However, the exact conditions under which perceptual decoupling occurs and its underlying cognitive mechanisms are still unclear. Hence, the present study systematically manipulated the task type (arithmetic, visuospatial) and workload (control, low, high) of the internal task in a within-subject design and tested its effects on voluntary saccades in a target-distractor saccade task. As expected, engagement in internal tasks delayed saccades to the target. This effect was moderated by time, task, and workload: The delay was largest right after internal task onset and then decreased, potentially reflecting the intensity of internal task demands. Saccades were also more delayed for the high compared to the low workload condition in the arithmetic task, whereas workload conditions had similarly high effects in the visuospatial task. Findings suggests that perceptual decoupling of eye behavior gradually increases with internal demands on general resources and that perceptual decoupling is specifically sensitive to internal demands on visuospatial resources. The latter may be mediated by interference due to eye behavior elicited by the internal task itself. Internal tasks did not affect the saccade latency-deviation trade-off, indicating that while the internal tasks delayed the execution of the saccade, the perception of the saccade stimuli and spatial planning of the saccade continued unaffected in parallel to the internal tasks. Together, these findings shed further light on the specific mechanisms underlying perceptual decoupling by suggesting that perceptual decoupling of eye behavior increases as internal demands on cognitive resources overlap more strongly with demands of the external task.

## Introduction

Many everyday cognitive tasks are performed independent of sensory input, like thinking about which route to a new shop is the fastest or summing up prizes during shopping. This internally directed cognition consumes cognitive resources that are shared with externally

---

**Data Availability Statement:** All materials, data, and code are publicly available at the Open Science Framework (https://osf.io/k4h92/) with DOI: 10.17605/OSF.IO/K4H92.

---

**Funding:** Work of SW and ZK was funded by the grant No. P34043 from the Austrian Science Fund (FWF). This grant was awarded to MB as principal investigator. The funders had no role in study design, data collection and analysis, decision to publish, or preparation of the manuscript.

**Competing interests:** The authors have declared that no competing interests exist.

directed cognition like perceiving and reacting to external visual input. Since both internally and externally directed cognition compete for similar cognitive resources, they interfere with each other when combined demands exceed available resources [1–3]. Hence, when we focus on internal activities like planning a route or mind wandering, fewer resources are available for external activities and we react slower, less accurately, or not at all to perceptual input (e.g., [4–6])–a phenomenon termed perceptual decoupling [7]. While perceptual decoupling gained quite some attention in mind wandering and attention research, the exact determinants and underlying cognitive mechanisms of this phenomenon are still unclear.

## Eye behavior during internally directed cognition

An easily assessable indicator of perceptual decoupling is eye behavior. When we focus on something in our surrounding, eye behavior is initiated to promote optimal perception: we direct our gaze to it, stabilize our gaze with microsaccades, adapt pupil size to luminance level, eye vergence to distance, and the pupil reacts to the intensity of processing of the perceptual input [8]. During internally directed cognition, eye behavior becomes less determined by external visual input–an indication of perceptual decoupling. For example, voluntary saccades become slower [9], the gaze is sometimes averted from visual input [10], microsaccade rate drops [11–13], pupil and vergence show activity unrelated to the visual input [14, 15], and the pupil's task-evoked response to the visual input decreases (e.g., [16]).

Internal activities themselves can also elicit changes in eye behavior [17, 18]. Research suggests eye behavior couples to aspects of mental representations. For example, the gaze is directed to previous, now empty positions of objects when trying to retrieve them [19], and mental arithmetic is associated with eye movements along the mental number stream [20].

## Limited shared resources as drivers of perceptual decoupling?

Whether and to what degree internally and externally directed cognition interfere with each other and eventually affect eye behavior seems to depend on the type of internally directed cognition. Internally directed cognition that involves visuospatial processes may interfere more with external visual tasks: For example, internal visuospatial tasks elicit more eye movements than verbal or arithmetic tasks [14, 21] and visual imagery reduces visual perceptual processing more than inner speech [22]. Prohibiting eye behavior or performing incongruent eye movements interferes with internal activities requiring spatial processes like remembering spatial relationships but not with internal activities requiring verbal or visual processes like remembering colors or names [20, 23–25]. Interference is larger for eye movements than mere shifts of attention [26], and not just eye movements but also limb movements interfere more with spatial than verbal tasks [27]. Hence, beyond visual and general attentional resources, visuospatial mental tasks seem to rely on spatial and motor planning resources that are also relevant to eye movements (see also [25]).

Besides task type, also the overall resource consumption (i.e., workload) should determine the occurrence and degree of perceptual decoupling. With increasing overall internal demands, fewer resources are available for externally directed cognition [1, 2] resulting in perceptual decoupling [28]. Hence, the interplay of type and workload of internally directed cognition should play an important role in perceptual decoupling. We confirmed this notion in a previous study that manipulated the type and workload of the internal task and investigated its effects on concurrent smooth pursuit eye movements [21]. Participants performed a numerical task (mental arithmetic, adding or subtracting) or a visuospatial task (mentally navigating through a matrix), and at the same time followed a moving dot on the screen. While mental arithmetic also involves some spatial processes as indicated the by beneficial effects of

congruent eye movements under certain conditions [29], it mainly draws from general and verbal/numerical resources (e.g., [30]); in contrast, mentally navigating through a matrix draws heavily on spatial resources [31], and similar tasks are even used to assess visuospatial capacities (e.g., symmetry span task; [32]). We found that, in the arithmetic task, workload showed a gradual effect on smooth pursuit eye movements: low workload caused little interference, and high workload caused more interference. Interestingly, in the visuospatial task, both low and workloads (supported by differences in pupil dilation) caused similarly strong interference with smooth pursuit eye movements. Exploratory analyses suggested a coupling of eye movements to the audio commands (up, down, left, right; [21]), which is consistent with the presumed interaction of task type and workload: The level of overall resource consumption gradually increases perceptual decoupling, but perceptual decoupling is most sensitive to the consumption of resources specifically required by the external perceptual task, in this case probably visuospatial processes for eye movement control.

In the present study, we aimed to replicate and extend these findings in the context of another well-established eye behavior under top-down control: saccades to a target vs. distractor. While smooth pursuit eye movements offer a continuous measure of perceptual decoupling, voluntary saccades offer additional insight into how internally directed cognition interferes with eye behavior: at the level of (a) perception and planning (reflected in the saccade latency-deviation trade-off), and/or (b) execution of eye movements (reflected in latency of saccades) [9]. Hence, we tested whether the effects of internal task type and workload on perceptual decoupling observed for smooth pursuit also extend to the planning and execution of voluntary saccades.

## Voluntary saccades

Saccades are the rapid movements of our eyes that align interesting objects (like possible prey, mates, or predators) in our visual environment with our fovea. The two-component framework for attentional deployment suggests that attention to objects in a scene is directed using both bottom-up and top-down cues [33]. A salient object can catch the eyes (bottom-up selection process), which competes with saccadic goals based on top-down selection processes (e.g., voluntary selection based on shape). Bottom-up selection processes are based on visual saliency, are pre-attentive, and fast (25 to 50 ms per item). Top-down selection processes depend on the task at hand (e.g., look at the square), and require voluntary effort, especially when faced with salient distractors that trigger the bottom-up selection process. This volitional directing of attention is controlled by higher areas and takes more time (at least 200 ms) [33]. With an increasing load of a secondary task, it is harder to inhibit reflexive saccades in a go/no-go saccade task, supporting the notion that the availability of general resources is required for the control of saccades [34].

A common task used to investigate voluntary saccades is the target-distractor saccade task [9]. When a target and a distractor are presented simultaneously, they both attract attention and the eye movement system starts planning saccades for both. Then, the system integrates information about the task and starts to change the saccade plans by inhibiting the eye movement plan to the distractor. When a saccade is executed before the distractor has been fully inhibited, the saccade trajectory will deviate towards the distractor [35, 36]. Hence, the deviation of the saccade per saccade latency (saccade latency-deviation trade-off) tells us about the state of spatial planning at the time the saccade is executed [9].

Reimer et al. [9] found that a parallel auditory-manual task delayed saccades in the target-distractor saccade task, but did not interfere with spatial planning of the saccade as indicated by no effects on the saccade latency-deviation trade-off. Interestingly, they further showed

that, whereas the go-condition of a go/no-go task delayed saccades, the no-go condition had no effect. This suggests that the manual response (button press in the go-condition) but not the response selection process (in both go and no-go conditions) of the auditory-manual task interfered with saccade execution [9]. What happens when an internal task is performed in parallel to the target-distractor saccade task?

## Current study

The present study aimed to replicate and extend our previous study on the effects of internal attention on smooth pursuit eye movements [21] by investigating the effects of task type and workload of internal tasks on voluntary saccades. In the numerical task, participants added or subtracted numbers, and in the visuospatial task, participants mentally moved a patch through a matrix [31]. Both internal tasks were performed with a low and high workload. In parallel to the internal task, participants performed the target-distractor saccade task: a target and a distractor were presented simultaneously on the screen and participants were asked to make a saccade to the target as fast as possible [9]. In our previous study, participants continuously performed smooth pursuit eye movements during the internal task. In the current study, the saccade task required participants to maintain fixation on the screen center, perceive the target and distractor, decide which one is the target, plan and execute a saccade there. The nature of eye movement (ballistic vs. continuous) and underlying mechanisms differ between the current and previous study. Hence, the current study tests whether effects of internal task performance found in the previous study also appear for voluntary saccades or show a distinct pattern, and investigate in more detail how the internal task interferes with perception, eye movement preparation and execution.

Our main research question was: Does voluntary saccade behavior serve as an index of internally directed cognition via the mechanism of perceptual decoupling? Specifically, is the execution of saccades to visual targets delayed when attention is turned inward (longer saccade latency)? If yes, is this effect modulated by the task type and/or workload (internal demands) of the internal task?

Based on previous studies and our research, we expected that the saccadic response in the target-distractor saccade task is delayed (longer saccade latencies) when performing a second task requiring internal attention, as indicated by a difference between dual-task (a parallel internal task with low and high workloads) and control condition (only target-distractor saccade task). We explored whether this effect is moderated by the task type (arithmetic vs. visuospatial) and/or the workload of the internal task. I.e., is the effect similar or different for the arithmetic and visuospatial task, and is the effect for the high workload condition (higher internal demands) larger than for the low workload condition (lower internal demands)?

We further explored the following non-preregistered questions: (1) Is maintaining fixation right before saccade target onset and (2) spatial planning of saccades to visual targets affected when attention is turned inward (change in saccade latency-deviation trade-off; [9])? And (3) given consistent eye behavior changes during internal tasks [14], we also examined to what degree is spontaneous eye behavior (i.e., blinks, spontaneous saccades) influenced by the internal tasks depending on the task type and workload of the internal task (Internal coupling)?

## Method

We provide our materials, data, and analysis scripts on the Open Science Framework (OSF, https://osf.io/k4h92/?view_only=7764e533530947d0a4d6b50f14897e8b). The study was pre-registered (AsPredicted #85098, https://aspredicted.org/7LJ_GNR). The minimal data set underlying the results and plots can be found here https://osf.io/47evd.

## Power analysis

We determined the sample size a priori based on a power analysis using G*power version 3.1 [37]. Based on our previous studies, we expected medium-sized within-subject effects of $dz = 0.4$ [14, 38, 39]. To have a power of 90% to detect an effect of 0.4 in the two-tailed pairwise *t*-tests, G*Power suggested a sample size of 44 participants. To account for possible exclusions of participants, we added ca. 15% to the sample size. Hence, we collected data from 50 participants.

## Ethics statement

The study protocol was approved by the Ethics Committee of the University of Graz and all participants gave written informed consent.

## Participants

Participants were pre-screened online via LimeSurvey based on the following criteria: normal (up to 0.5 diopters) or corrected to normal vision (soft contact lenses only), native German speaker, no dyslexia, no dyscalculia, no problems distinguishing left and right, no neurological or psychological disorders, no eye sicknesses, no previous eye surgery affecting vision, no active medication affecting eyesight or driving abilities. All participants gave written informed consent and were paid 10 € per hour or received partial course credit. Personal information (signed informed consent and info for reimbursement and course credit) was saved separately from any data reported in this study, is only accessible by the authors, and gets deleted following university's data security rules.

Data were collected between January to March 2022. The local COVID-19 regulations in this period required participants and experimenters to wear FFP-2 masks, have a "Green Pass" (at least two vaccinations, one vaccination and one recovery, or negative PCR-Test within the last 24 h), and fill out a COVID-19 screening form.

Data from 49 participants were analyzed. One additional participant had to be excluded due to very low task performance in the saccade task (below 20% correct, see below) which indicated a misunderstanding of instructions. The 49 participants (35 female) were aged between 19 and 35 years ($M = 23.51$, $SD = 2.27$). All participants were university students, 16 already held a university degree, and 37 had normal vision and 12 wore soft contact lenses.

## Tasks, design, and procedure

In the present study, participants performed a dual-task paradigm with an internal task, requiring them to repeatedly perform a mental operation [21], and an external task, requiring them to make a saccade to a target and ignore a distractor (Fig 1). The main experimental manipulations of this dual-task paradigm were (1) that the internal task type was either arithmetic or visuospatial, (2) that the workload of the internal task was either zero (control condition = only external task), low, or high, and (3) that the time between the onset of the internal task operation and the onset of the saccade target for the external task (Stimulus Onset Asynchrony; SOA) was either 0.5, 1, 1.5, 2, or 2.5s. These manipulations allowed us to assess whether and how task type and workload of internal tasks affected performance in the saccade task over time, in specific, latency and accuracy of saccades to the target. We return to the specifics of the tasks and manipulations in the corresponding sections below.

## Internal tasks

As in previous work [21, 40], we used two different internal tasks: a mental arithmetic task and a visuospatial task, and three levels of workload (control, low, high). In the *arithmetic task*,

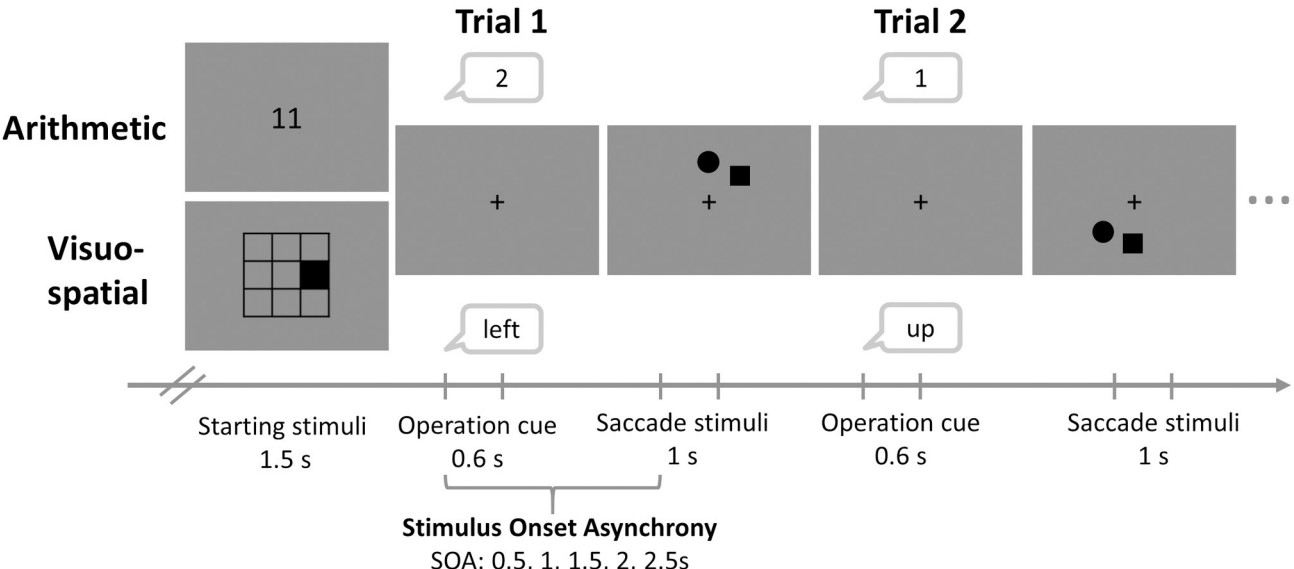

**Fig 1. Time course of trials within a task block (arithmetic or visuospatial task).** Each block started with information on the workload of the block (control, low, high). After a drift check and a 2s fixation cross (not depicted here), the starting stimulus of the internal task appeared for 1.5s followed by a fixation cross which remained on the screen for the rest of the block. The first operation cue was auditorily presented 2s after the offset of the starting stimulus. The onset of an operation marked the onset of a trial and the end of the previous trial. At a given time after operation onset (SOA of 0.5s to 2.5s, with steps of .5s), the saccade stimuli (a circle as the target and a square as the distractor, or vice versa) were presented for 1s followed by a variable inter-trial-interval. The total trial duration varied from 3.6s to 4.4s in steps of 0.1s. At the end of each block, participants were asked to report the result of the internal task (this report is used as the performance measure).

participants had to mentally add or subtract a number. The task was based on [41]. At the beginning of each block, a starting number appeared on the screen for 1.5s, followed by a sequence of 9–11 operations, i.e., a number from 1 to 4 was presented via speakers which had to be continuously added to or subtracted from the intermediary result (Fig 1). Time between operations varied between 3.6s and 4.4s in steps of 0.1s. In the blocks of the *low workload* condition, the starting number was between 10 and 40, and the presented numbers 1 or 2 had to be added. In the blocks of the high worklo*ad* condition, the starting number was between 65 and 99, and the presented numbers 3 or 4 had to be subtracted. The starting numbers were chosen so that the results were between 10 and 100. In the blocks of the control condition, "XX" appeared instead of a starting number and, the same numbers were presented auditorily as in the dual-task conditions (half from low and half from high load condition), but no internal task had to be performed. The sequence of numbers was randomized. At the end of each arithmetic task block (after 9–11 trials), participants had to type in the result. In the control condition, they had to leave the response field empty.

In the *visuospatial task*, participants had to mentally navigate through a matrix (similar to Kerr, 1993). At the beginning of each block, a matrix with one black patch was shown, indicating the starting location (Fig 1). Then a sequence of 9–11 operations (i.e., statements of direction) was presented via speakers (in German) indicating in which direction the patch should be moved mentally (up, down, left, right). The time between operations varied from 3.6s to 4.4s in steps of 0.1s. Different levels of workload were realized by using a 3 x 3 matrix (ca. 2.7 x 2.7 dva) in the *low workload* condition and a 4 x 4 matrix (ca. 3.6 x 3.6 dva) in the high load condition. The sequence of operations was randomized with the restriction that the patch did not leave the matrix and that a current operation does not undo the previous operation (because that is mentally less demanding). In the blocks of the control condition, an empty

matrix (3 x 3 and 4 x 4 in half of the blocks, respectively) and the same operations were presented as in the dual-task conditions, but no internal task had to be performed. At the end of each visuospatial task block (after 9–11 trials), an empty matrix appeared and participants had to report the last location of the patch by clicking on the respective cell of the matrix. In the control conditions, they were instructed to leave the matrix empty.

Each block of the arithmetic and visuospatial task started with brief information about the workload condition (control, low, high). Participants continued with pressing the spacebar, the eye-tracker performed a drift check followed by a fixation cross (2s) and the presentation of the starting stimulus for the respective internal task (1.5s). Then, the respective internal task started (see Fig 1).

In case participants lost track of the arithmetic or visuospatial task during a block, they were instructed to resume the task with the last remembered number/location and report this result. In case they were completely unable to perform the internal task in a block (e.g., due to mind wandering or external distraction), they should leave the response field empty, so we could exclude this block from analyses. The workload conditions of the internal tasks were tested in pilot tests and settings were chosen to differ in perceived difficulty and performance but being neither too easy nor too demanding.

For each internal task (arithmetic, visuospatial), there were 30 blocks, 10 per workload condition (control, low, high). Each block comprised 9 to 11 trials defined by one internal operation (number to add/subtract or direction to move the patch mentally). Hence there were 300 trials per internal task. The order of operations within a block was set quasi-randomly but was the same for all participants. The order of blocks within an internal task was randomized between participants.

## External task: Voluntary saccades

As an external task, participants performed a saccade task in parallel to the internal tasks. During each internal task trial, participants performed one saccade. Participants were asked to keep their eyes on a black fixation cross (0.18 x 0.18 dva) in the center of the screen (Fig 1). A saccade target, and a distractor were presented simultaneously for 1 s after each auditory operation cue in the internal task. Time intervals between operation onset and saccade target onset varied between 0.5 and 2.5s (SOA = 0.5, 1, 1.5, 2, 2.5, each interval was used 10 times, order randomized). This range of SOAs was chosen to cover phases of intense internal focus during the task performance (directly after audio around 0.5 and 1s) but also the phase when the internal manipulation is already over (around 2–2.5s). For half of the participants, the saccade target was a black circle with a 1.12 dva diameter and the distractor was a black square with 1 x 1 dva, whereas the opposite assignment was used for the other half of the participants. Target and distractor were presented 8 dva from the center of the screen at one of 8 principal positions (steps of 45˚) on an imaginary circle around the fixation cross, with an angle of 45˚ between them (hence the distance between the center of the target and the center of the distractor was 6.12 dva). Participants were instructed to make a saccade to the target as fast as possible and return to the fixation cross afterward.

## General procedure

Upon arrival in the lab, participants gave informed consent and we confirmed their eyesight with the Landolt vision test [42]. They filled out a questionnaire regarding their current physical and mental state (i.e., coffee, alcohol consumption, sleep, vigilance).

Participants performed the dual-task paradigm for both internal tasks (arithmetic and visuospatial) with counterbalanced task order across participants. Participants received

instructions for the tasks on paper and screen. After participants understood the task, they were seated in front of the eye tracker and they practiced the task in four practice blocks (two control, one low, and one high workload condition). Participants had to give correct responses at the end of each practice block of the internal task to continue with the main blocks, or else the practice block was repeated.

With an average of 10 operation and saccade stimuli pairs per block, 5 different SOAs, and 10 blocks per workload condition (control, low, high), there were 20 trials per SOA, task, and workload combination per participant.

After completion of each internal task (arithmetic and visuospatial), participants evaluated perceived demand, fun, distraction, exhaustion, and vigilance, and were asked to indicate whether they used specific strategies to perform the tasks. In the end, they reported motivation to perform well, which task they prioritized (internal, external, or both), perceived duration of the lab session, and perceived appropriateness of the number of breaks (see supplemental materials).

Overall, the dual-task paradigm lasted around 1.5h, ca. 45 min per internal task and participants were allowed to take breaks between tasks. Within the same lab session, participants also performed another paradigm with a different research question, half before and half after the current paradigm.

## Apparatus

The eye-tracking session took place in a sound-attenuated room without daylight and lights on (luminance at the participant's place was 29.55 lx). Participants were seated in front of a 24-inch ASUS VG248qe monitor (1920 x 1080 pixels, ca. 33.52˚ x 19.73˚, 60 Hz, monitor settings: 40% brightness, 20% contrast).

Binocular eye-behavior was tracked at 1,000 Hz with an EyeLink 1000 Plus system (SR Research Ltd.). A chin rest stabilized the participant's head and kept a distance to the screen at 88 cm. The eye tracker with the illuminator was placed 59 cm in front of the chin rest. A 9-point calibration and validation were performed at the beginning of each task (arithmetic, and visuospatial) with the thresholds of average gaze error kept below 0.5˚ and maximum error below 1˚. Drift checks were conducted at the beginning of each task block. If drift check failed, a recalibration was performed.

All letters and numbers were black (0.37 dva high, font "Arial") and the background was grey (RGB 128,128,128). Auditory operations were presented through a Logitech PC speaker Z 200 with computer volume at 100% and speaker volume at medium. Audio files were created using https://wideo.co/text-to-speech/ with voice "[de-DE] Lisa Fischer-S", converted to Ogg files, for compatibility with PsychoPy, and edited to an equal length of 600ms. The experimental script was generated in PsychoPy (Version 2020.2.10; [43]).

## Data preprocessing

Eye tracking data were retrieved via EyeLink Data Viewer Software (SR Research Ltd., version 4.2.1) and further processed with R (R Version 4.2.2, [44]) using RStudio (Version 2022.12.0, [45]). Blinks were detected by the eye tracking software and extended by 100 ms backward and forward to account for possible data distortions due to partial lid closure. Gaze position and pupil diameter data were preprocessed separately.

Pupil diameter was transformed from arbitrary units to millimeters following instructions from SR Research, smoothed with a moving average filter (n = 20), and interpolated during blinks using the *gazeR* package [46]. We excluded samples with data from only one eye, with abnormal eye vergence (gaze position difference of left and right eye greater than the inter-

pupil distance of 60 mm) fixation disparity outliers (beyond 3 SD from individuals mean), with abnormal pupil diameter data (beyond natural limits of 2–8 mm), pupil outliers (beyond 3 SD from individuals mean), and samples within saccades (average excluded samples per participant: arithmetic task: 3.77%, visuospatial task: 3.86%). On the trial level, a total of 808 trials (2.69%) of the pupil data were excluded (due to 50% or more missing data per trial, and no response to the internal task), leaving pupil data from a total of 29,192 trials for analyses.

For analyses of tonic pupil diameter as a measure of overall arousal [47], median pupil diameter in the first 500ms after operation cue onset was used, a phase in which participants were required to maintain fixation. For analyses of phasic pupil diameter (task-evoked pupil response TEPR) as a measure of within-task changes in attentional effort [47], we binned pupil data from operation cue onset until the onset of the saccade stimuli into 500ms time windows and baseline corrected them by subtracting pupil diameter in the 500ms before operation onset [48]. Presentation of the saccade target itself and especially the gaze position change when making a saccade to the target causes changes in pupil diameter [49] and therefore pupil data after the onset of the saccade target were excluded from pupil diameter analyses. Hence, later time bins include fewer data (i.e., only trials with an SOA of 2.5s provide pupil data for the last time window 2–2.5s).

Gaze position samples were excluded when they were within blinks, fixation disparity was abnormal or an outlier, pupil diameter was abnormal or an outlier (see above), and when only data from one eye were available (average excluded samples per participant: arithmetic task: 8.84%, visuospatial task: 8.52%). Samples were categorized as saccade if velocity was larger than 30˚/s or acceleration was larger than 8000˚/s$^2$ and if this event lasted more than 6ms.

For analyses of saccades in the saccade task (landing position, latency), we excluded trials with no saccade, with more than 50% missing gaze position data, trials with no response to the internal task, when gaze was not on the fixation cross at saccade onset as instructed (gaze more than 2 dva from screen center), and when saccade latency was shorter than 80 ms or longer than 600 ms [9]. This led to a total exclusion of 3,908 trials (13.03%) and left 26,092 trials (86.97%) for further analysis. S1 Table provides the number and proportion of excluded/included trials for each criterion and task type and workload combination.

We analyzed the first saccade made after the saccade target onset. Saccades were categorized as landing on-target or on-distractor when the landing position was within 3 dva from the target center or distractor center, respectively. The criterion of 3 dva was determined based on the distribution of the distance of the first saccade's landing position to the target and differs from preregistration (1.68 dva). From all included trials, in 18,619 cases (71.36%) first saccades landed on the target, in 5,653 trials (21.67%) on the distractor, and in 1,820 trials (6.98%) somewhere else.

### Analysis strategy

For internal task performance, we calculated binomial GLMMs, using the response per block (correct, incorrect). Participants reported the result of the arithmetic task, and the final position of the patch in the matrix, respectively, at the end of a block, hence after 9–11 trials, leading to one performance measure per block, not trial, see Fig 1.

For subjective ratings we calculated 2x2 repeated-measures ANOVAs with the two within-subject factors Task (arithmetic vs. visuospatial) and Workload (low vs. high) (control task had per se no "task performance" and no rating) and used repeated-measures pairwise *t*-tests for planned pairwise comparisons.

Pupil diameter and saccade latency data were analyzed with linear mixed models (LMMs), and external task performance data, i.e., maintained fixation, and saccade on target, were

analyzed with binomial generalized mixed models (GLMMs) using the *lme4* [50] and *lmerTest* package [51], with the Satterthwaite's method to calculate degrees of freedom. Each model included the fixed factors Task (arithmetic vs. visuospatial), Workload (control, low, high), and Time (for pupil: 500ms time bins from 0.5 to 2.5s; for saccade latency: SOAs from 0.5 to 2.5s), and their interactions (Task x Workload x Time) as well as consecutive block number, time of paradigm (first of second) and time of task (first or second) as a fixed covariate (non-significant covariates were removed from the final model), and random intercepts for participant and trial (trial ID). Global fixed effects for each model were tested with a Type III ANOVA using the function implemented into the *lmerTest* package. For the follow-up planned pairwise comparisons, we used the *emmeans* package [52] instead of pairwise t-tests as preregistered because *emmeans* uses the fitted model as input and therefore produces better estimates of the data. We provide effect sizes using an approximation to Cohen's *d* computed with the *eff_size* function from *emmeans*, which relies on the model's residual degrees of freedom and estimated population *SD*.

Besides the estimate of the magnitude of effects (Cohen's *dz*), we also report Bayes Factors (BFs) as an estimate of the weight of evidence in favor or against our hypotheses. BFs were computed with the *BayesFactor* package [53] under a default Cauchy prior for each model and pairwise comparison based on the data aggregated across trials. BFs below three can be interpreted as weak, between 3 and 20 as moderate, between 20 and 150 as strong, and larger than 150 as very strong evidence [54]. We interpreted effects that had a *p*-value below 0.01 (significant) and a BF10's larger than 3 (weight of evidence in favor of hypothesis). For descriptive statistics and plots, we used the *summarySEwithin* function from the *Rmisc* package [55] which adjusts confidence intervals for within-subject designs using the method from [56].

## Results

### Internal tasks

We first wanted to confirm the effectiveness of the workload manipulation in the internal tasks. We analyzed perceived demand (Fig 2A) and pupil diameter (baseline and task-evoked) (Fig 2C and 2D) as measures of subjective and objective workload, and proportion correct responses (Fig 2B) as measures of net task performance.

### Perceived demand

Participants perceived the internal tasks in the high workload condition (arithmetic: $M = 3.84$, $SD = 0.89$; visuospatial: $M = 4.16$, $SD = 0.94$) as more demanding than in the low workload condition (arithmetic: $M = 3.18$, $SD = 0.95$; visuospatial: $M = 3.04$, $SD = 0.78$) ($F_{1,48} = 60.51$, $p < .001$, $dz = 0.12$, BF10 = 18,806.18). See Fig 2A. This effect was slightly larger in the visuospatial task than in the arithmetic task (interaction Task*Workload: $F_{1,48} = 6.63$, $p = .013$, $dz = 0.01$, BF10 = 0.48). The main effect of the Task was not significant ($F_{1,48} = 0.27$, $p = .607$, $dz < 0.01$, BF10 = 0.17).

### Task performance

We tried to calculate binomial GLMMs for internal task performance, using the response per block (correct, incorrect). (Participants reported the result of the arithmetic task, and the final position of the patch in the matrix, respectively, at the end of a block, hence after 9–11 trials, leading to one performance measure per block, not trial, see Fig 1). However, the models did not converge and had singularity. We therefore reverted to a repeated-measures ANOVA using the proportion of correct blocks per Task and Workload.

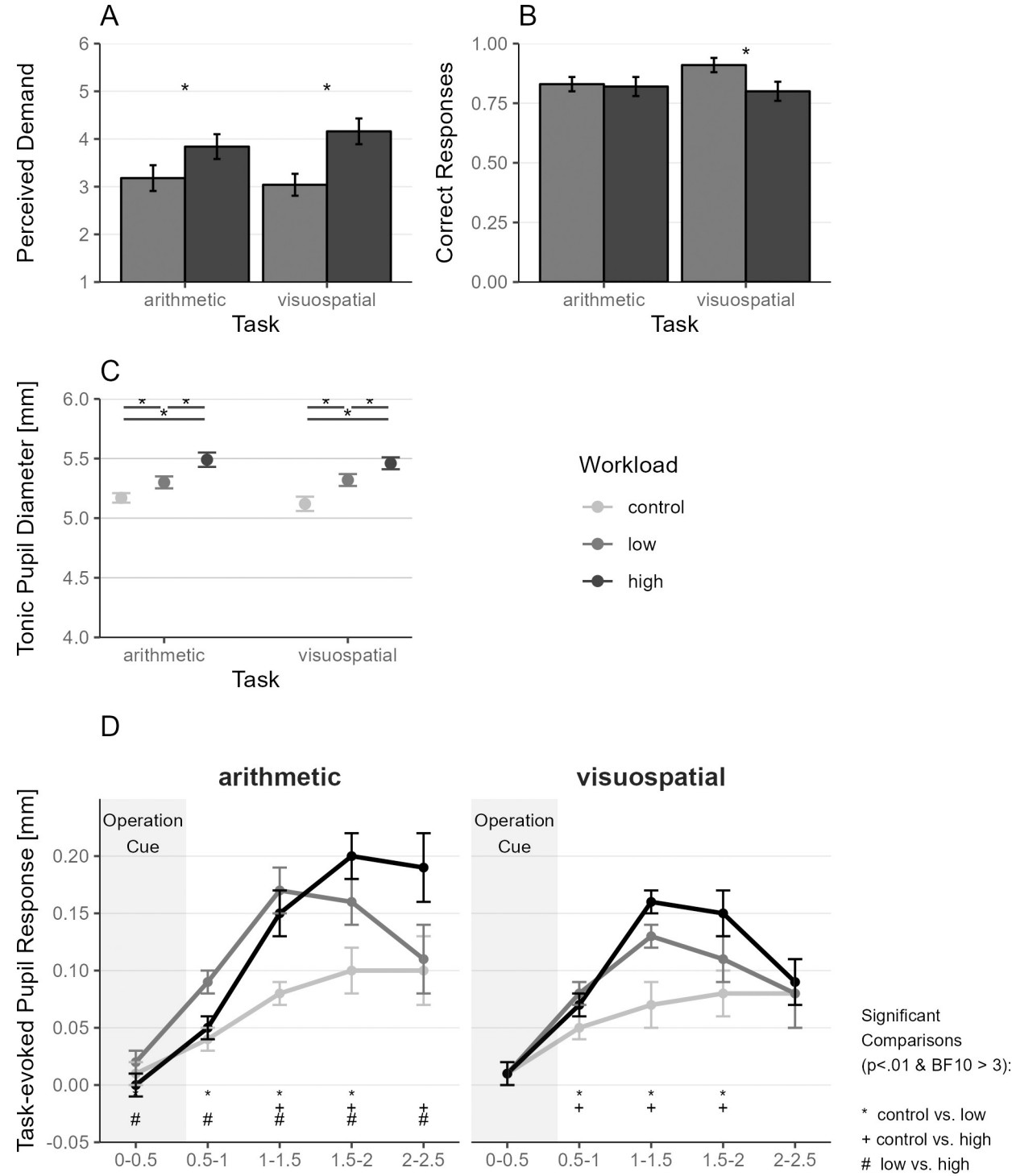

**Fig 2.** Internal Task Performance: Perceived Demand (A), Correct Responses (B), Tonic Pupil Diameter (C), and Task-evoked Pupil Response (D). (A) Perceived demand of the low and high workload condition of the internal tasks, reported after completion of each paradigm. (B) The proportion of blocks with correct responses in the internal tasks (participants had to report the final result after 9–11 trials at the end of a block, see Fig 1). (C) Average pupil diameter in the first 500ms of a trial, which reflects the overall arousal level induced by the workload conditions [47]. Asterisk indicates differences between workload conditions at $p < .01$ and BF10 > 3. (D) Change of pupil diameter in response to the task = Task-evoked pupil response. This measure of phasic pupil diameter reflects task-evoked changes in the intensity of attention/workload [47]. Values indicate the mean and error bars indicate the 95% confidence interval (adjusted for within-subject designs using the method from [56]) Shaded area indicates operation cue presentation (0–0.6s). $N$ = 49.

Regarding performance in the internal tasks, we found a significant main effect for Workload ($F_{1,48} = 11.18$, $p = .002$, $dz = 0.04$, BF10 = 7.44) and a significant (but BF10 <3) interaction of Task*Workload for the proportion of correct responses ($F_{1,48} = 9.46$, $p = .003$, $dz = 0.03$, BF10 = 2.93; see Fig 2B) in the repeated-measures ANOVA. The main effect Task was not significant ($F_{1,48} = 2.57$, $p = .116$, $dz = 0.01$, BF10 = 0.38). In the visuospatial task, performance was better in the low compared to the high workload condition (low: $M = 0.91$, $SD = 0.10$; high: $M = 0.80$, $SD = 0.13$; $t_{48} = 4.81$, $p < .001$, $dz = 0.17$, BF10 = 1289.65). In the arithmetic task, performance did not differ significantly between low and high workload conditions (low: $M = 0.83$, $SD = 0.11$; high: $M = 0.82$, $SD = 0.15$; $t_{48} = 0.44$, $p = .664$, $dz = 0.02$, BF10 = 0.17).

## Tonic/Baseline pupil diameter

To see how the internal task and workload conditions affected the overall level of arousal, we analyzed tonic/baseline pupil diameter [16, 47, 57]. We calculated a linear mixed effects model predicting tonic pupil diameter (first 500ms of the trial see section 2.5. Data preprocessing) with the fixed effects Task (arithmetic, visuospatial) and Workload (control, low, high) and their two-way interaction, as well as Block, Task Order, and Time of Paradigm (not sign.) as a covariate to control for time on task effects. Main effect Task ($F_{1,29090} = 26$, $p < .001$, BF10 = 0.19) and Workload ($F_{2,29091} = 1,444.5$, $p < .001$, BF10 > 100,000) and their interaction ($F_{2,29090} = 23.2$, $p < .001$, BF10 = 1.43) predicted tonic pupil diameter (note, BF10 < 3). See S2 Table for random and fixed factors.

In both internal tasks, the workload conditions were reflected in baseline pupil diameter in the expected order: control, low, and high workload, with the control showing the smallest and high showing the largest baseline pupil diameter (Fig 2C). We observed no difference in baseline pupil diameter between the arithmetic and the visuospatial tasks ($t \leq 6.56$, $p > .0001$, BF ≤ 0.84). Hence, the manipulation of workload was successful in terms of tonic arousal and allocated attentional resources [47]. Further, baseline pupil diameter decreased linearly with blocks ($F_{1,29090} = 195.0$, $p < .001$, BF10 > 100,000) and slightly with task order ($F_{1,29090} = 31.5$, $p < .001$, BF10 = 0.57), which likely reflects practice effects or vigilance decrement [47].

## Task-evoked pupillary response (TEPR)

Next, we tested how the task type and workload manipulation affected the within-trial allocation of attentional effort by assessing the task-evoked pupillary response (TEPR) [47] to the operation cues. For that, we calculated a linear mixed effects model predicting baseline-corrected pupil diameter with the fixed effects Task, Workload, and Time and their interactions as well as block number, task order and time of paradigm. Results and pairwise comparisons are in Fig 2D, Table 1 and S4 Table, and random and fixed effects in S3 Table.

**Table 1. Task-evoked Pupillary response (TEPR): Type III ANOVA table for the LMM model showing fixed effects.**

| Effect | $DF_{num}$ | $DF_{den}$ | F | p | BF10 | BF01 |
|---|---|---|---|---|---|---|
| Task | 1 | 87279.86 | 108.65 | < .001 | 657.22 | < 0.01 |
| Workload | 2 | 86641.82 | 306.59 | < .001 | > 100,000 | < 0.01 |
| Time | 4 | 87283 | 1125.96 | < .001 | > 100,000 | < 0.01 |
| Task*Workload | 2 | 86864.87 | 3.11 | 0.045 | 0.04 | 25.91 |
| Task*Time | 4 | 87300.01 | 31.52 | < .001 | > 100,000 | < 0.01 |
| Workload*Time | 8 | 87275.35 | 59.32 | < .001 | > 100,000 | < 0.01 |
| Task*Workload*Time | 8 | 87283.41 | 8.62 | < .001 | 124.62 | 0.01 |

TEPR = Task-evoked pupil response. Using Satterthwaite's method. BF = Bayes Factor; BF10 = ratio of evidence in favor of effect; BF01 = ratio of evidence against the effect. N = 49, total observations = 87,380.

While the overall arousal levels reflected in tonic/baseline pupil diameter had similar patterns for the arithmetic and visuospatial task, the tasks slightly differed in their task-evoked pupillary responses. Both internal tasks (at low and high workloads) elicited an increase in pupil diameter compared to the single-task control condition. In the visuospatial task, both low and high workloads increased pupil diameter up until time bin 1–1.5s, and then pupil diameter leveled and decreased again. Hence, moving a patch in a 3x3 or a 4x4 matrix was associated with a similar degree of attentional effort for a similar time despite differences in overall arousal (tonic pupil diameter). In the arithmetic task, the low workload condition also led to an increase in pupil diameter up to the time point 1–1.5s; yet, the high workload condition further increased pupil diameter up to the time point 1.5-2s, reflecting higher and longer attentional effort for subtracting than adding. Please note here, that although pupil diameter reacts to the attentional effort, this reaction is slow and therefore lags behind the actual peak of attentional effort (which is within the first 1.5s). Together, these results provide further support for the effective manipulation of workload conditions.

## External task

The external task—a target-distractor saccade task simultaneous to the internal tasks—required participants to maintain fixation on a fixation cross and make a saccade to an appearing target and inhibit saccades toward the simultaneously appearing distractor. We tested how performance in this saccade task was affected by the type of the internal Task (arithmetic vs. visuospatial) and Workload (control, low, high) across different SOAs. Below, we report effects on the following measures that reflect different aspects of external task performance: maintaining fixation before target onset, correct saccade to the target, and latency of saccade to the target (main dependent variable of interest). Additionally, we test if effects on the proportion of correct saccades to the target are a consequence of effects on saccade latency or whether our manipulations affected spatial planning of saccades (saccade latency-deviation trade-off).

## Maintaining fixation

How well participants were able to maintain their gaze on the fixation cross right before the saccade to the target of the target-distractor saccade task offers a first indication of how the internal tasks affect performance in an external task requiring eye behavior. We calculated a binomial generalized mixed model (GLMER) with successful maintained fixation (gaze within 2dva from screen center at onset of the saccade to the target, yes = 1, no = 0) as the dependent binomial variable and Task, Workload, and Interaction of Task*Workload as predictors. We included block number, task order and time of paradigm as covariates, but only task order contributed to the model.

The main effect of Workload was significant ($\chi^2_2 = 31.75$, p < .001, BF10 > 100,000), with a lower proportion of maintained fixation in the high workload condition compared to the low and control conditions in both the arithmetic and visuospatial task (Fig 3A, S6 Table). Main effect Task ($\chi^2_1 = 1.7$, p = .193, BF10 = 0.25) and the interaction Task*Workload were not significant ($\chi^2_7 = 2.05$, p = .358, BF10 = 0.06), see also random and fixed effects in S5 Table. Including SOA as a predictor in the ANOVA did not yield any significant effect for SOA ($\chi^2_4 = 7.39$, p = .117, BF10 = 243.75) or interactions with SOA ($\chi^2_{4-8} > = 12.95$, p > = .114, BF10 > = 0.01). These findings indicate that a higher workload resulted in less reliable fixation right before the saccade.

## Correct saccades to target

Next, we wanted to know, how Task and Workload conditions affected the accuracy of saccades to the target across different SOAs. Correct saccade to the target was defined as the first

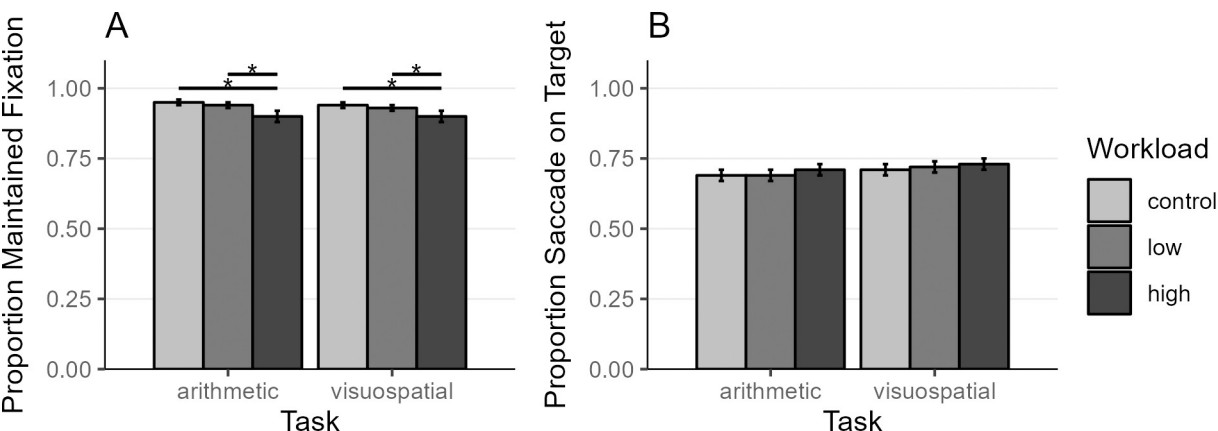

**Fig 3.** External Task Performance: Proportion of Trials with Maintained Fixation (A), and Saccade on Target (B). Participants were instructed to maintain their gaze on the fixation cross during a trial and then make a saccade to the appearing saccade target and inhibit saccades toward the simultaneously appearing distractor. (A) The proportion of trials in which participants successfully maintained fixation on the fixation cross (gaze within 2 dva from the center of fixation cross) at the onset of the saccade to the target. (B) The proportion of trials in which the first saccade after the saccade target onset landed on the saccade target (gaze within 3 dva from saccade target center). The asterisk indicates differences between workload conditions with $p < .01$ and $BF10 > 3$. Values indicate mean and error bars indicate 95% confidence interval (adjusted for within-subject designs using the method from [56]). $N = 49$.

saccade after saccade target onset, landing on the target (gaze within 3 dva from saccade target center). Only trials were considered in which fixation had been maintained right before saccade target onset. There was a main effect for Workload ($\chi^2_2 = 23.81$, $p < .001$, $BF10 = 18.91$), and significant interaction of Workload*SOA ($\chi^2_8 = 47.37$, $p < .001$, $BF10 = 6.58$) and Task*-Workload*SOA ($\chi^2_8 = 110.73$, $p < .001$, $BF10 > 100,000$). Main effect Task, SOA and the interactions Task*Workload and Task*SOA were not significant ($\chi^2_{1-8} \leq 26.68$, $p \geq .001$, $BF10 \leq 1.05$) (see S7 Table for fixed and random effects). Follow-up pairwise $t$-tests showed a complex pattern (see S8 Table). Overall, participants tended to make more successful saccades to the target with increasing internal workload (Fig 3B). Since saccade latency and saccade accuracy are intertwined through the saccade latency-deviation trade-off, these results need to be interpreted considering this latency-deviation trade-off (see Fig 4C).

### Saccade latency

In the next step, we analyzed the effects of task, workload, and SOA on saccade latency, which is the central measure in the saccade task. The distribution of saccade latencies per Task and Workload are presented in Fig 4A. The linear mixed effects model and follow-up comparisons showed that saccades to targets were delayed in both internal tasks compared to the control condition, with effects being strongest in the first SOA and getting smaller for later SOAs (see Fig 4B, Table 2 and S9 and S10 Tables).

Pairwise task comparisons (S10 Table) and Fig 4B showed that the arithmetic and the visuo-spatial tasks had similar effects on saccade latency in the high workload condition with prolonged saccade latencies for short SOAs and then a decrease in saccade latency and a leveling off above the control condition. In the low workload condition, the visuospatial task showed an increase in saccade latency for the first SOA bin and leveled above the control condition similar to the high workload condition. The low load arithmetic task showed a smaller increase in saccade latency than high load arithmetic task for the first two SOAs and then dropped to the level of the control condition. Differences between low and high workload conditions were higher in the arithmetic task than in the visuospatial task. Further, saccade latency decreased over time (Time on Block), potentially reflecting practice effects (see Table 2).

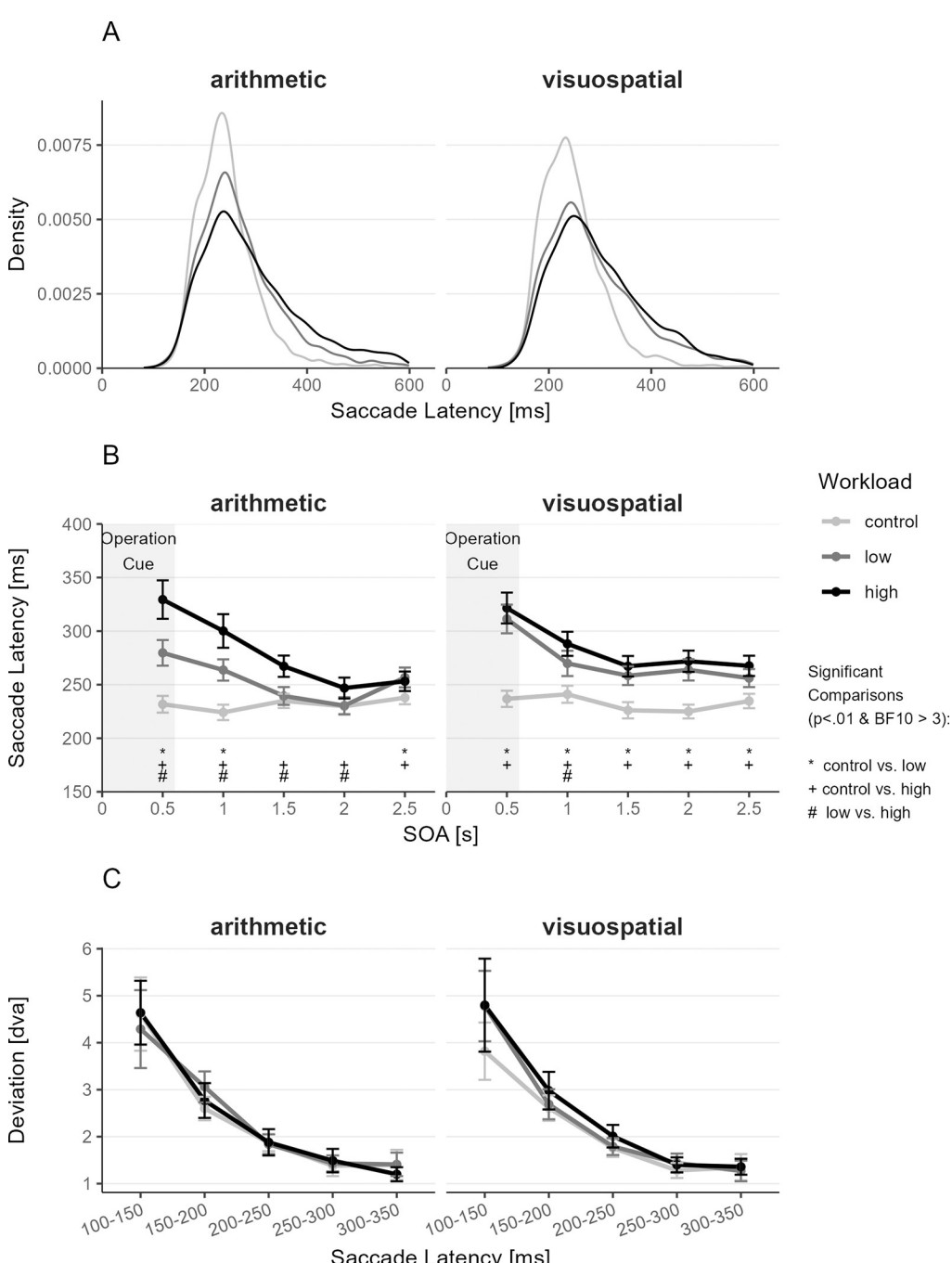

**Fig 4.** Distribution of Saccade Latencies (A), Saccade Latency per SOA (B), and Saccade Latency-Deviation Trade-Off (C), each depicted separately for both internal tasks as a function of workload condition. (A) Distribution of saccade latencies of the first saccade after saccade stimuli onset. (B) Latency of first saccades after saccade stimuli onset, landing within 3 dva from the target center. (C) Relation between saccade latency of the first saccade after saccade stimuli onset and distance between the center of the target and the landing position of the saccade (i.e., saccade latency-deviation trade-off; Reimer et al., 2021). Dots indicate the mean, and error bars indicate the 95% confidence interval (adjusted for within-subject designs using the method from [56]). Differences with p < .01 and BF10 ≥ 3 are marked with * for control vs. low; with + for control vs. high; and with # for low vs. high. Significance was defined as $p < .01$ and BF10 > 3. $N = 49$.

**Table 2. Saccade latency: Type III ANOVA table for the LMM model showing fixed effects.**

| Effect | $DF_{num}$ | $DF_{den}$ | F | p | BF10 | BF01 |
|---|---|---|---|---|---|---|
| Task | 1 | 18533.35 | 35.35 | < .001 | 10042.31 | < 0.01 |
| Workload | 2 | 18532.64 | 759.29 | < .001 | > 100,000 | < 0.01 |
| Time (SOA) | 4 | 18537.76 | 163.22 | < .001 | > 100,000 | < 0.01 |
| Time of Block | 1 | 18530.92 | 72.57 | < .001 | > 100,000 | < 0.01 |
| Task Order | 1 | 18531.89 | 297.02 | < .001 | > 100,000 | < 0.01 |
| Task*Workload | 2 | 18530.97 | 17.53 | < .001 | 34456.19 | < 0.01 |
| Task*Time | 4 | 18537.9 | 1.73 | 0.141 | < 0.01 | 1775.81 |
| Workload*Time | 8 | 18538.59 | 38.71 | < .001 | > 100,000 | < 0.01 |
| Task*Workload*Time | 8 | 18533.94 | 7.57 | < .001 | 9004.61 | < 0.01 |

Time = refers to the SOA (Stimulus Onset Asynchrony between trial onset and saccade task onset); Using Satterthwaite's method. BF = Bayes Factor; BF10 = ratio of evidence in favor of effect; BF01 = ratio of evidence against the effect. N = 49, total observations = 18,619.

## Saccade latency-deviation trade-off

As saccade latency and proportion of saccades on target were affected by Task and Workload, we next checked the saccade latency-deviation trade-off (Fig 4C), to see whether internal demand affected the spatial planning of saccades [9]. To this end, we ran an LMM with the distance of the landing position to the target center as the dependent variable and binned saccade latency (150–200, 200–250, 250–300, 300–350 ms), Task (arithmetic, visuospatial), and Workload (control, low, high), and SOA as independent variables. The general saccade latency-deviation trade-off was present in our data as reflected in the significant main effect of saccade latency ($F_{3,21665.19} = 372$, $p < .001$, BF10 > 100,000): Increased saccade latency was associated with higher precision in saccade execution. These results suggest the observed increase in saccade accuracy with a higher workload (see Fig 3B) is likely explained by an increase in saccade latency. All interactions with saccade latency were not significant ($F \leq 1.96$, $p \geq .023$, BF10 < 0.01) (see S11 Table). The absence of interaction effects indicates that the saccade latency-deviation trade-off was not affected by task type or workload conditions, and thus that internal demands did not affect the spatial planning of saccades.

## Exploratory analyses: Disengagement and internal coupling

Above we found that the internal Task and Workload conditions interfered with maintaining fixation and prolonged saccade latencies to an external target. Research on internally directed cognition showed that internal tasks themselves can elicit changes in eye behavior (e.g., increased blinks and saccadic activity; 13). Changes in eye behavior due to disengagement (e.g., increase in blinks) and internal coupling (e.g., increase in spontaneous saccades) might partly explain the observed effects of poorer fixation and increased saccade latency during internal tasks. Hence, in the next step, we assessed whether blink and saccade activity were increased during internal tasks compared to the control condition. For that, we divided the time from operation cue onset to saccade target onset into five 500ms time windows (similar to the analysis of task-evoked pupil response above) and computed the number of blinks and saccades.

Finally, we checked whether the observed effects of the internal tasks on saccade latency can be attributed to the eye behavior changes elicited by the internal task (in the 500ms right before the saccade target onset).

**Blinks.** Internal tasks are consistently associated with higher blink rates than external tasks, associated with visual disengagement during internal tasks and/or increased visual

preparedness in external tasks [14, 58–60]. We calculated a generalized mixed effects model predicting whether a blink occurred or not with the fixed effects Task (arithmetic, visuospatial), Workload (control, low, high), time bin (time since operation cue onset), and their interactions. We included the time of block, task order and time of paradigm as a covariate (time on task effects) and random intercepts for participants and trial. The Type III ANOVA test of the global fixed effects showed significant effects for Workload ($\chi^2_2$ = 9.54, p = .008, BF10 = 22,487.08), Time ($\chi^2_4$ = 106.5, p < .001, BF10 > 100,000) and the Interaction of Workload and Time ($\chi^2_8$ = 76.99, p < .001, BF10 = 3.31). Task, and the other interactions were not significant or did not have BF10 > 3 ($\chi^2_{1-8}$ ≤ 18.42, $p$ ≥ .018, BF10 ≤ 0.04; see Fig 5A). The interaction of Workload and Time was explored in pairwise $t$-tests (see S12 Table). The blink rate was at a similar level at the beginning of the trial for all workload conditions, but low and high workload conditions (internal task) had a relatively higher blink rate than the control condition at about 1–1.5s after trial onset.

**Saccades (pre-target).** Internal tasks can also elicit saccadic activity associated with internal task demands compared to external tasks that often require a restriction of eye behavior like maintaining fixation before the saccade target onset [14]. We calculated a generalized mixed effects model predicting whether a saccade occurred or not with the fixed effects Task (arithmetic, visuospatial), Workload (control, low, high), time bin (time since operation cue onset), and their interactions. We included the time of block as a covariate (time on task effects) and random intercepts for participants and trial. The Type III ANOVA test of the global fixed effects showed the following effects: Task ($\chi^2_1$ = 7.87, p = .005, BF10 > 100,000), Workload ($\chi^2_2$ = 16.68, p < .001, BF10 > 100,000), Time Bin ($\chi^2_4$ = 11.31, p = .023, BF10 > 100,000), the two-way interaction of Task*Workload ($\chi^2_2$ = 11.01, p = .004, BF10 = 0.37), Task*Time Bin ($\chi^2_4$ = 4.51, p = .0341, BF10 = 0.19) and Workload*Time Bin ($\chi^2_8$ = 50.92, p < .001, BF10 = 3.37), and the three-way interaction of Task*Workload*Time Bin ($\chi^2_8$ = 44.54, p < .001, BF10 = 0.19; see Fig 5B). Follow-up pairwise $t$-tests can be found in S14 Table. The saccade rate was at a similar level for all conditions at the beginning of the trial. The saccade rate then increased for the low and high Workload conditions (internal task) compared to the control condition. This increase in saccade rate was earlier and more pronounced in the visuospatial task (time bin 0.5-1s) than in the arithmetic task (time bin 1–1.5s). These findings suggest that executing mental operations increased saccadic activity. In the control condition, the saccade rate was relatively stable across the trial. Hence, merely hearing numbers or spatial directions ("up", "down", "left", "right") did not automatically increase spontaneous saccade activity. Please note that the overall number of saccades before the saccade task onset was generally low, hence detailed analyses of direction of those saccades was not possible due to low power (e.g., arithmetic task high load: only around 153 trials per direction, visuospatial task high load: only around 226 trials per direction from all participants combined). Nonetheless, for interested readers, we included a plot of saccade directions in the S1 Fig. S1 Fig does not suggest any systematic covariation of saccade directions relative to task cues (e.g., addition vs. subtraction) of those pre-saccade task saccades as previously reported (e.g., [61]). Hence, it is possible that typical spontaneous eye movements following the internal number stream or movement through the matrix, respectively, were suppressed when preparing to perform the saccade task.

**Blinks and saccades pre target as predictors of saccade latency.** The analyses above showed that the internal tasks elicited changes in eye behavior (increased blink and saccade rates) compared to the control condition. So, next, we checked whether these changes were associated with the previously observed effects of the internal tasks on saccade latency in the saccade task. To this end, we reran the model of saccade latency (section 3.2.4) with the additional fixed effects of blink and saccade rate in the 500 ms right before the saccade target onset.

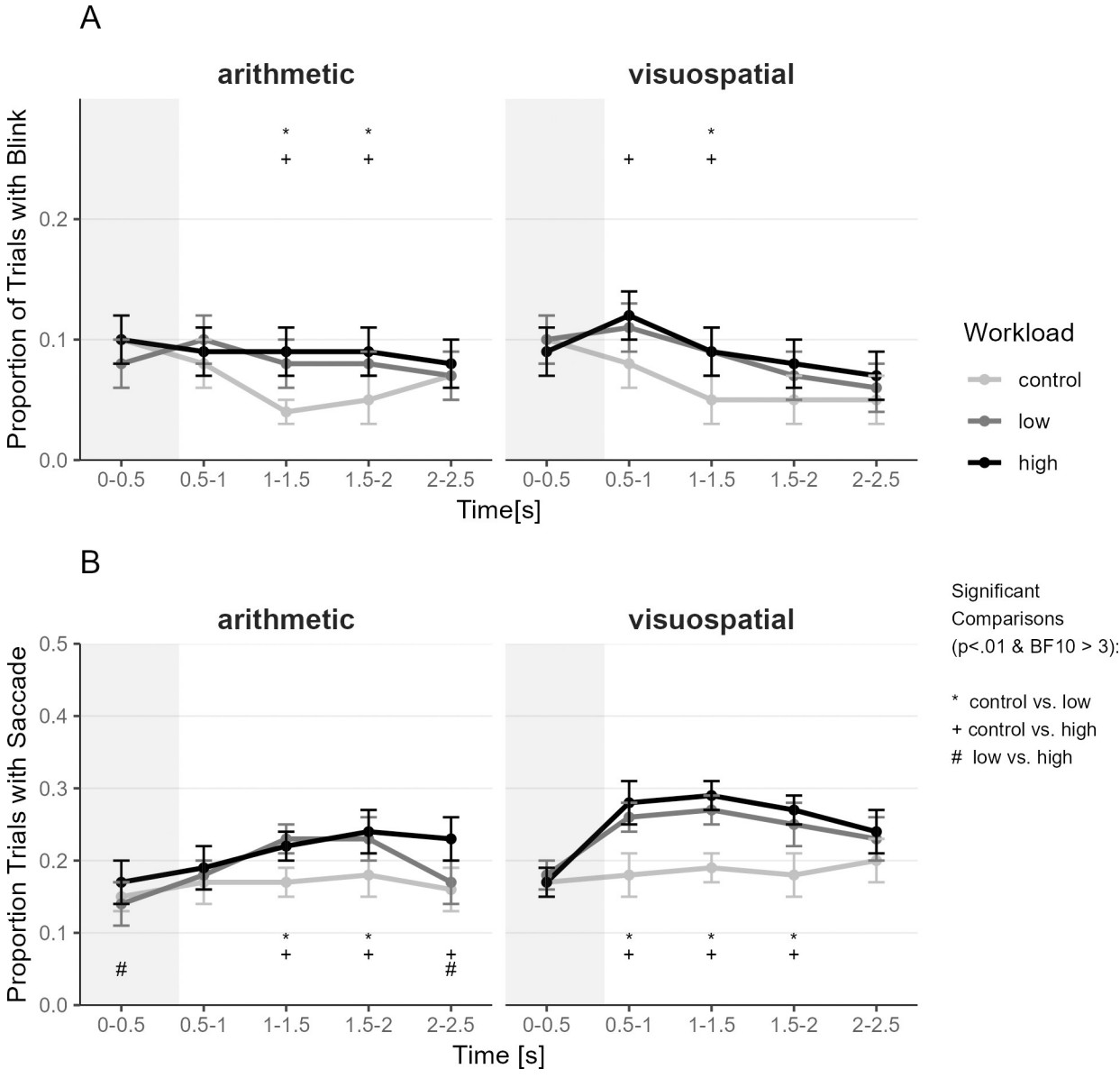

**Fig 5.** Blinks (A) and Saccades (B) across Trial (before saccade stimuli onset) as a function of Internal Task and Workload. (A) Blinks and (B) Saccades per second (Hz) were binned into 500 ms time windows. Only time windows that occurred before a saccade target onset are analyzed because saccade target presentation would affect blinking and saccades. Hence, later time windows include only data from trials with longer SOAs. Dots indicate the mean, and error bars indicate the 95% confidence interval (adjusted for within-subject designs using the method from [56]). Workload differences with p < .01 and BF10 > 3 at given time points are marked with * for control vs. low; with + for control vs. high; and with # for low vs. high. *N* = 49.

Model comparison showed that including blinks and saccades as predictors increased model fit ($\chi^2_2 = 330$, p < .001). The results of the new model are shown in Table 3 (and S16 Table). Saccades and blinks before the saccade onset prolonged the latency of saccades to the target, with the effect being small for saccades and large for blinks. Nonetheless, controlling for saccades and blinks only marginally altered all the earlier reported effects. Hence, the effects of task type and workload reported in section 3.2.4 are not the only consequence of eye behavior changes elicited by the internal tasks.

**Table 3. Extended model of saccade latency with pre-target saccades and blinks as additional predictors: Type III ANOVA table for the LMM model showing fixed effects.**

| Effect | $DF_{num}$ | $DF_{den}$ | F | p | BF10 | BF01 |
|---|---|---|---|---|---|---|
| Saccades | 1 | 18573.17 | 11.79 | 0.001 | 0.45 | 2.23 |
| Blinks | 1 | 18569.26 | 318.97 | < .001 | > 100,000 | < 0.01 |
| Task | 1 | 18531.59 | 38.61 | < .001 | > 100,000 | < 0.01 |
| Workload | 2 | 18530.87 | 747.6 | < .001 | > 100,000 | < 0.01 |
| Time | 4 | 18535.58 | 156.59 | < .001 | > 100,000 | < 0.01 |
| Time of Block | 1 | 18528.87 | 73.41 | < .001 | > 100,000 | < 0.01 |
| Task Order | 1 | 18530.01 | 293.6 | < .001 | > 100,000 | < 0.01 |
| Task*Workload | 2 | 18528.97 | 17.42 | < .001 | 9699.92 | < 0.01 |
| Task*Time | 4 | 18536.05 | 1.78 | 0.13 | < 0.01 | 2762.7 |
| Workload*Time | 8 | 18536.45 | 40.31 | < .001 | > 100,000 | < 0.01 |
| Task*Workload*Time | 8 | 18531.18 | 7.61 | < .001 | > 100,000 | < 0.01 |

Using Satterthwaite's method. BF = Bayes Factor; BF10 = ratio of evidence in favor of effect; BF01 = ratio of evidence against the effect. N = 49, total observations = 18,619.

## Discussion

The phenomenon of perceptual decoupling describes the reduced processing and response to perceptual input during internally directed cognition [7]. While more and more studies investigate perceptual decoupling, the exact determinants of perceptual decoupling during internally directed cognition remain unclear. The present study systematically tested the effects of task type (arithmetic, visuospatial) and workload (control, low, high) of the internal task on the execution of voluntary saccades in a target-distractor saccade task. Varying the time between the internal task operation cues and the saccade task stimuli further allowed us to investigate the temporal dynamics of perceptual decoupling. We found that the internal tasks delayed saccades to targets and this effect was moderated by the type, workload, and time dynamics of the internal task. While internal tasks delayed the execution of saccades, spatial planning of saccades continued without impairment, resulting in delayed but slightly more accurate saccades to targets. Exploratory analyses further revealed additional effects of internal task performances on eye behavior independent of the saccade task. Together, our findings elucidate the role of general and specific resource consumption (i.e., task type and workload, respectively) for the degree of perceptual decoupling.

The current study aimed to conceptually replicate and extend our previous investigation of internal attention effects on smooth pursuit eye movements [21]. Both studies used the same manipulation of type and workload of the internal task but differed in the eye behavior addressed by the external task. The target-distractor saccade task used in this study allowed us to test whether perceptual decoupling effects observed by [21] extend to a different form of top-down eye behavior and to additionally distinguish between effects on eye movement planning and execution [9].

As in our previous study, workload manipulations were clearly reflected in baseline pupil diameter differences, with the pupil diameter increasing from control to low to high workload conditions. Tonic pupil diameter was similar for arithmetic and visuospatial task, suggesting that tasks elicited similar levels of arousal [47]. Closer inspection of the task-evoked pupil response showed that tasks still differed in the magnitude and duration of invested attentional effort within the trial [47]., as mental subtraction (high load arithmetic task) required slightly more and longer-lasting effort than mental addition (low load) or moving a patch in a mental

matrix (low and high load visuospatial task). In the arithmetic task, the increase in task-evoked pupil diameter from time bin 0–0.5 to 0.5-1s was slightly less steep in the high compared to the low load condition, although the high load condition then caught up and exceeded the low condition. In the arithmetic task, participants had to switch from addition to subtraction for the high load condition, maybe this caused the delay in pupil diameter increase. Taken together, internal tasks successfully elicited increased workload demands.

Internal tasks had pronounced effects on the voluntary saccade task. The saccade task required participants to maintain fixation on a fixation cross and to quickly execute a saccade to an appearing target vs. distractor. The internal tasks already impaired the first part of the saccade task: With increasing workload of the internal tasks, participants were less able to maintain fixation right before the saccade stimuli onset. Hence, performing an internal task even interfered with the relatively simple task of maintaining fixation.

Regarding the execution of saccades to the target, we observed an interaction between task type and workload. The effect of the arithmetic task on saccade latencies was moderated by the workload condition (low vs. high), consistent with the workload effects observed for task-evoked pupil diameter: the high load condition delayed voluntary eye movements more and for a longer period than the low load condition. In the visuospatial task, although within-task allocation of attentional effort (task-evoked pupil response) was similar to the low load condition of the arithmetic task, both low and high workload conditions of the visuospatial task delayed voluntary eye movements as strongly as the high load condition of the arithmetic task. These findings replicate those by [21] where low and high workload conditions of the visuospatial task differed in baseline pupil diameter but both strongly and equally impaired voluntary eye behavior in the smooth pursuit task. These impairments of voluntary saccades by internal attention demands are consistent with the perceptual decoupling hypothesis. Stronger effects of the visuospatial task are likely attributed to the fact that a visuospatial mental task shares more modality-specific resources with voluntary eye movements than the arithmetic task–suggesting an important role of shared resources for the degree of perceptual decoupling.

Moreover, the degree of impairment of voluntary eye movements depended on the time course of the internal task performance as studied by SOAs. Whereas saccade latencies were delayed only for early SOAs in the arithmetic task, saccade latencies were elevated across all SOAs in the visuospatial task. This suggests that the arithmetic task impaired voluntary eye movements only during performing the arithmetic operation but not during maintaining the interim result in working memory, while the visuospatial task impaired voluntary eye movements during both stages, performance of the mental navigation and maintenance of the end position in working memory. It seems that performing the arithmetic operation (but not retaining a number) shares some general resources with the planning and execution of eye movements. In contrast, navigating through an imaginary matrix and remembering that spatial position utilizes more specific resources that are also required for the planning and execution of eye movements. Hence, the latter task lead to stronger interference [21]. These findings again indicate that perceptual decoupling is sensitive to the modality of the internal task, with more pronounced perceptual decoupling caused by visuospatial tasks due to higher modality-specific interference.

We may still ask why in the visuospatial task workload effects were observed for baseline pupil diameter and fixation maintenance but not for task-evoked pupil diameter and saccade latency? Mentally moving a patch in a matrix might imply an increase in overall arousal due to increased cognitive effort but the same resources are employed within the task regardless of the size of the matrix. Alternatively, it is also possible that there is a threshold for shared resources or a ceiling effect for perceptual decoupling: there is a gradual increase in interference (as in the arithmetic task) but as soon as a certain number of essential resources are

bound by the internal task, the impairment of voluntary eye movements reaches a maximum. Further studies are required to explore this relationship between the amount of shared resources and the impairment of voluntary eye movements in more detail.

So far, this study replicated and added further support for the stronger interference between eye behavior and spatial tasks compared to numeric or verbal tasks [20, 22–25], and for the interplay of task type and workload as determinants of perceptual decoupling [21].

In our previous study [21], participants had to perform smooth pursuit throughout the trial. In the current study, the saccade task required participants to make a decision which object is the target and then execute the saccade there. Hence, the saccade task was more demanding and required a switch of attention from the internal task to the external task. Switching attention between internal and external focus is costly [62]. Could it be that participants first completed the internal task and then performed the saccade task? To achieve the saccade latencies in the current study under the assumption that participants first finished the internal tasks, they would have to finish the internal task in only 0.55 to 0.6 s (target onset at 0.5s + latency–latency of control condition) after operation cue onset, hence already before the end of the audio operation cue. To avoid artefacts of manual responses, we did not assess how long participants needed to perform the internal operations in the present study. However, we had ran pilots in our lab during the planning phase of the paradigms, in which we had assessed response times in a self-paced version of some variations of the internal tasks (without an additional external task). Our pilot participants needed around 1.9 s for the arithmetic task in the high load condition (median = 1.896, SE = 0.143, N = 20), and around 0.8 s for the visuospatial task (low load: median = 0.775, SE = 0.069, N = 12; high load: median = 0.850, SE = 0.072, N = 21). Hence, it seems unlikely that participants first completed the internal task before performing the saccade task. This interpretation is further supported by the fact that saccade latencies were still increased after the first SOA. The available evidence rather supports the idea that participants interrupted the internal task, switched to the external task and then resumed the internal task. Intense focus on the internal task might have delayed perception of the saccade target and the switch of tasks itself caused delays due to shared resources [62].

In the current study, participants heard the same audio operations in the control condition as in the experimental conditions in order to control for effects of just hearing the audio on saccade latency. Simply hearing the audio could already altered the saccade latency in the saccade task, hence we have no pure measure of how long the saccade latency would be in the saccade task when there is no other paradigm at the same time. In the control condition, participants did receive no starting number (XX instead) and only an empty matrix, and we explicitly asked them to not perform any internal task. It is still possible that they tried to perform the internal task, e.g., out of boredom. However, we see no increase in saccade latency for the first SOAs in the control condition and also task-evoked pupil diameter in the control condition did not show the same pattern as in the low and high load condition. Hence, the data suggests that participants followed instructions and did not perform any internal task during the control condition. Nonetheless, further studies should also involve a control condition without the audios to get an estimate of the saccade latency in a complete single task condition.

The current study extends previous work offering additional insights on how internal tasks specifically interfere with voluntary eye movements. Compared to the smooth pursuit task, the target-distractor saccade task allows us to distinguish between the process of eye movement planning and the process of eye movement execution [9]. We found that the internal tasks interfered with saccade execution (prolonged saccade latency) but not with saccade planning which was found to continue unaffectedly during the internal task (no change in saccade latency-deviation trade-off). [9] discovered a similar effect of an auditory-manual task on the

target-distractor saccade task: delayed saccade execution but unaffected spatial planning (reflected in no change in saccade latency-deviation trade-off). They further showed that the go-condition but not the no-go condition of a go-no-go task delayed saccade execution, and therefore argued that the manual response but not the response selection process of the auditory-manual task interfered with saccade execution [9].

The internal tasks in our study did not require any explicit motor response. However, previous studies showed that internal tasks can elicit eye movements ("internal coupling"; e.g., [14, 21, 63], and eye movements are thought to play a functional role in internally directed cognition (E.g., memory reinstatement; [24, 29]). In the present study, performing the internal task made it harder to maintain fixation right before the saccade stimulus onset, and the visuospatial task and–to a lesser degree–the arithmetic task were accompanied by an increase in saccades before the saccade stimuli onset, suggesting some "internal task-related" eye movements. An additional analysis showed that saccadic activity–potentially triggered by the internal tasks (saccades before the onset of the saccade target and distractor)–partially explained the delay in saccade execution in the saccade task. Hence, the involuntary planning and/or executing of eye movements coupled to the internal tasks could have interfered with executing voluntary saccades in the target-distractor saccade task in a similar way as the button presses did in the study [9]. In line with that, [26] showed that eye movements and limb movements can have similar effects on verbal and spatial tasks, suggesting that the motor aspect and not the visual aspect of eye movements is likely the source of interference. Hence, the extent of motor responses elicited by the internal tasks could be another determinant of the degree of perceptual decoupling. For further support, a future study could include a task involving limb movements but no eye movements and compare it to the present tasks.

The internal tasks did not affect the spatial planning of the saccade to the target as indicated by no effects of the internal tasks on the saccade latency-deviation trade-off. In the study [9] with an auditory-manual task, there were also only effects on saccade execution but not on spatial planning of the saccade. Hence, spatial planning of a saccade to a target seems to continue in parallel to the performance of mental arithmetic, mental spatial navigation, or an auditory-manual task [9] without interference, suggesting little or no overlap in resource consumption. Also, an increase in workload did not affect the saccade latency-deviation trade-off in the present study, suggesting that spatial planning of saccades was independent of both general and modality-specific resource consumption of the internal tasks. Further studies are needed to replicate this effect and test whether it also holds for other internal tasks.

## Conclusion

To sum up, using the target-distractor saccade task, this study provided new insights on what aspects of an internal task interfere with what aspects of voluntary eye movements, shedding further light on the determinants of perceptual decoupling. We can conclude that (1) perceptual decoupling is sensitive not just to the amount but also the modality of the mental resources consumed by an internal task, (2) the eye movements associated with internal activities appear partly responsible for the modality-specific interference, and (3) the interference happens at the level of eye movement execution but probably not at spatial planning.

## Supporting information

**S1 Table. Excluded and included trials per internal task and workload condition.**
(DOCX)

**S2 Table. Baseline pupil diameter: Random and fixed effects.**
(DOCX)

**S3 Table. Task-evoked pupillary response (TEPR): Random and fixed effects.**
(DOCX)

**S4 Table. Pupil diameter TEPR: Pairwise comparisons for the interaction of task and workload per time bin.**
(DOCX)

**S5 Table. Maintained fixation: Random and fixed effects.**
(DOCX)

**S6 Table. Maintained fixation: Pairwise comparisons of workload per task.**
(DOCX)

**S7 Table. Correct saccade to target: Random and fixed effects.**
(DOCX)

**S8 Table. Correct saccade to target: Pairwise comparisons of task and workload per SOA.**
(DOCX)

**S9 Table. Saccade latency: Random and fixed effects.**
(DOCX)

**S10 Table. Saccade latency: Pairwise comparisons of workload and task per SOA.**
(DOCX)

**S11 Table. Saccade latency–deviation trade-off: Random and fixed effects.**
(DOCX)

**S12 Table. Blinks: Random and fixed effects.**
(DOCX)

**S13 Table. Blinks: Pairwise comparisons of workload per task and time bin.**
(DOCX)

**S14 Table. Saccades before saccade target: Random and fixed effects.**
(DOCX)

**S15 Table. Saccades before saccade target: Pairwise comparisons of workload per time bin.**
(DOCX)

**S16 Table. Extended model of saccade latency: Random and fixed effects.**
(DOCX)

**S1 Fig. Visualization of the direction of saccades that occurred prior to saccade target onset.** (A) Effect of the arithmetic task on the direction of pre-target saccades. X-axis shows the principal direction of the saccade, y-axis the proportion of all pre-target saccades that were in this direction, and color whether it was addition or subtraction and whether it was control or internal task condition (low, high workload). (B) Proportion of pre-target saccades that were made in the direction of the audio command. X-axis shows the direction that was presented via headphones, y-axis shows the proportion of all pre-target saccades that were made in the direction of the audio, and color shows the workload condition (control, low, high). Please note that number of pre-target saccades was overall low.
(TIF)

## Acknowledgments

We thank Ivo Ohling and Naomi Ernst for their help with data collection.

## Author Contributions

**Conceptualization:** Sonja Walcher, Živa Korda, Christof Körner, Mathias Benedek.

**Data curation:** Sonja Walcher, Živa Korda.

**Formal analysis:** Sonja Walcher.

**Funding acquisition:** Mathias Benedek.

**Investigation:** Živa Korda.

**Methodology:** Sonja Walcher, Christof Körner.

**Project administration:** Sonja Walcher, Mathias Benedek.

**Software:** Sonja Walcher.

**Supervision:** Christof Körner, Mathias Benedek.

**Visualization:** Sonja Walcher.

**Writing – original draft:** Sonja Walcher.

**Writing – review & editing:** Sonja Walcher, Živa Korda, Christof Körner, Mathias Benedek.

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
