## [Decision Letter · Decision Letter 0]

12 May 2023

PONE-D-23-07956The effects of type and workload of internal tasks on voluntary saccades in a target-distractor saccade taskPLOS ONE

Dear Dr. Walcher,

Thank you for submitting your manuscript to PLOS ONE. After careful consideration, we feel that it has merit but does not fully meet PLOS ONE’s publication criteria as it currently stands. Therefore, we invite you to submit a revised version of the manuscript that addresses the points raised during the review process.

We look forward to receiving your revised manuscript.

Kind regards,

Jie Wang, Ph.D.

Academic Editor

PLOS ONE

Journal Requirements:

"We thank Ivo Ohling and Naomi Ernst for their help with data collection. This work was supported by the Austrian Science Fund (FWF): P34043."

"Work of SW and ZK was funded by the grant No. P34043 from the Austrian Science Fund (FWF). This grant was awarded to MB as principal investigator.

3. We noted in your submission details that a portion of your manuscript may have been presented or published elsewhere. [The results of this study will be presented as poster at the Conference "Current Issues in Mind-Wandering Research - Theoretical Advances and New Empirical Findings" in Heidelberg July 2023 and as part of a talk at the Conference "European Conference on Visual Perception" in Cyprus August 2023. Only the abstract will appear in the corresponding proceedings.] Please clarify whether this [conference proceeding or publication] was peer-reviewed and formally published. If this work was previously peer-reviewed and published, in the cover letter please provide the reason that this work does not constitute dual publication and should be included in the current manuscript.

Additional Editor Comments:

The authors should pay careful attention to each of the comments below and address the issues raised by the three reviewers.

Reviewers' comments:

Reviewer's Responses to Questions

**Comments to the Author**

1. Is the manuscript technically sound, and do the data support the conclusions?

Reviewer #1: Yes

Reviewer #2: Yes

Reviewer #3: Yes

2. Has the statistical analysis been performed appropriately and rigorously? 

Reviewer #1: Yes

Reviewer #2: Yes

Reviewer #3: Yes

3. Have the authors made all data underlying the findings in their manuscript fully available?

Reviewer #1: Yes

Reviewer #2: Yes

Reviewer #3: Yes

4. Is the manuscript presented in an intelligible fashion and written in standard English?

Reviewer #1: Yes

Reviewer #2: Yes

Reviewer #3: Yes

5. Review Comments to the Author

Reviewer #1: This is a nice, preregistered, eye-movement study exploring the interplay between mental workload and saccades. Participants were asked to complete a mental task (arithmetic or visuo-spatial) with different degrees of load (zero, low, high) and then to perform a saccade towards a target while ignoring a distractor. To sum up, the main results show an influence of the mental task on eye movements, and in particular this was moderate by temporal parameters (SOA), and the type of workload.

I think this paper is clear and well organised. I only have some comments in the hope that these could help the authors.

My main comment is about statistics. It is relatively common to see papers reporting both frequentist and Bayesian analyses, such as in this case. As you performed power analyses, the main results are those concerning the frequentist framework; bayes factors are surely a nice addition, but I am feeling that some readers could be confused; for example, on page 16, you reported that you considered as ‘significant’ a result with p < .001 (I think here you mean .01?) and BF10 > 3. Although the concept of “significance” sounds good for p values, I believe it could be misleading for BF (they actually provide evidence for H0 or H1, depending on the direction). You could think to reframe this part a bit, and the following passages where you refer to statistical ‘significance’.

Another comment is related to methods: On page 13, you say that participants completed another experiment with a different purpose; I can see the practical reasons for that, but I would like to know if all participants completed first the task reported here and then the other task; if not, could tasks order may have played a role?

Other minor comments:

Page 3, paragraph 1.1: I think that, in addition to the citations (11 and 12), it would be fair and relevant to cite two additional papers showing a link between working memory load and microsaccades (Dalmaso et al., 2017).

Page 9: you used a relatively wide range of SOAs; I would appreciate to see some more words explaining this choice and if this was based on previous studies.

References

Dalmaso, M., Castelli, L., Scatturin, P., & Galfano, G. (2017). Working memory load modulates microsaccadic rate. Journal of Vision, 17(3), 6. https://doi.org/10.1167/17.3.6

Reviewer #2: The authors' study centers around perceptual decoupling during internally directed tasks, with a particular focus on measuring the effects of voluntary saccades on task type and workload. This work builds upon and replicates earlier research by the same authors, with the goal of examining whether voluntary saccade behavior can serve as an indicator of internally directed cognition through the mechanism of perceptual decoupling.

The methods utilized in this study appear to be sound, and the research itself seems to be conducted with rigor. However, there are a few questions that need to be addressed before publication can be granted.

1)

It would be beneficial for the authors to highlight the potential novelty of their study in the introduction, going beyond the goal of simple replication. One way to accomplish this could be to explore whether there are specific reasons why perceptual decoupling with voluntary saccades might manifest differently than in previous reports.

For instance, it may be worth considering the ballistic nature of saccades, which distinguishes them from other types of movements such as manual pointing and reaching. This qualitative difference could potentially lead to distinct patterns in perceptual decoupling, and could further support the originality of the study. By emphasizing these potential novel aspects of their research, the authors can more effectively justify their work and add to the existing body of knowledge in the field.

2)

The authors utilized a within-participant experimental design where each participant was involved in all experimental conditions. In the control condition, participants were presented with identical numeric or visuo-spatial input, but without a clear starting point. Specifically, participants received an XX character in the numeric condition and an empty matrix in the visuo-spatial condition.

However, it is important to note that due to transfer effects from the main tasks, participants may have attempted to perform internal operations based on the numeric and visuo-spatial input, even though they were not explicitly instructed to do so. If this were the case, it is possible that a partial and uncontrolled form of perceptual decoupling could have occurred during the control condition as well.

It is important for the authors to consider this potential scenario and its implications for their study. They may want to discuss it in their results section and explore potential ways to control for this confounding factor in future research.

3)

The quality of the figures appears to be inadequate. It is recommended to generate figures at a higher resolution to enhance their clarity.

Reviewer #3: It was a pleasure to read this manuscript. The authors present excellent and comprehensive work on how internal tasks (mental arithmetic and spatial planning) interfere with externally directed gaze behaviour. The manuscript is clearly written and showcases outstanding scientific quality in adopting open-science standards and adhering to highly commended power calculation. I just have a few points that probably deserves being discussed in more detail and that might require additional analyses.

Major points

- The main conclusion seems to be that perceptual decoupling delays the saccades in the external task. Although I agree that the pattern or results is consistent with this explanation, I believe that drawing this conclusion is perhaps a bit too pre-mature. The authors show nicely that eye movements associated with internal activities contribute to the delayed saccades effect, but cannot explain it completely. (There is one minor comment about this model, see below.) Thus, there is some residual delay in the saccade latency which, as the authors infer, must be due to perceptual decoupling. Couldn’t it also be that participants, in each trial, conducted first the internal task operation and once this process was finished, the saccade was executed. With a higher load, the internal task operation took more time and therefore, there was a longer delay until the saccade. This explanation would suggest that internal and external tasks were more or less executed one after the other; whereas the perceptual decoupling explanation rather suggests that the saccade target was not perceived that well because participants were internally doing something else, which is a bit different. Even if there are reasons against this sequential tasks explanation, I recommend to discuss it.

- The pupillary response measured with a video-based eye-tracker is per definition affected by the gaze position on screen. The current experimental design balanced the saccade target locations, it is thus expected that any such influence cancels more or less out. However, since quite some trials were excluded, saccade target locations are not balanced anymore in the dataset used in the analysis. Did the authors consider that? see e.g. Gagl et al. (2011) in Behav. Res. Methods for how to deal with this.

- General remark on the mixed models: I have the impression that R’s default contrasts were used. They make interaction effects hard to interpret. I recommend to use effect coding or successive difference contrasts, because they make interaction effects interpretable as with a default Anova. I’m not sure, though, whether the anova table with type III SS takes care of that. If it does, no worries.

- I recommend to report also the models’ random effects structures together with the fixed effects output, not only the Anova tables, perhaps in the supplement.

- (Just a suggestion because perhaps of interest, if not already known, to the authors.) In the absence of visual input, gaze behaviour seems to reflect a mental number line for numerical tasks. Eyes deviate towards left for smaller and towards right for larger numbers (e.g. Loetscher et al., 2010, Curr. Biol). The authors might want to see whether the addition and subtraction tasks show a similar effect in the the internal-task-related eye movements before saccade target onset.

Minor points

- p. 7 typo “; 9)?”

- p. 13 Was there really no re-calibration at any point within a task block?

- section 2.6 Why a 2x2 ANOVA with workload as 3-level factor? In addition, this line sounds like the analysis here was on proportion correct/error responses. It would have been more consistent to run a mixed model on single trial data not only for saccade latency and pupil but also for task performance (i.e. a binomial model with logit link or similar).

- section 2.6. paragraph 2: What was the random intercept for “trial” based on? Was it on something like a trial number, i.e. somehow corresponding to time within the experiment, or was it based on the stimulus item of a trial, thus rather a trial ID, e.g. all trials with saccade target at location 12 o’clock coded with the same trial ID. In any case, trial ID would be the option that makes sense.

- p. 16 ““significant” if the p-value is below 0.001 and the BF10’s is larger than 3.” This statement is surprising and makes p-values and BF conclusions rather sound incompatible. A p-value of 0.001 is usually associated more with a BF > 20 or similar. I recommend to either stick to p 0.001 and BF 20 or 0.01 and BF 3, or something like that. I see later, e.g. Fig. 2 uses p < .01. Perhaps the earlier .001 was a typo.

- Fig 2D arithmetic: looks like pupil response started later in the high compared to low and control conditions. Any post-hoc idea why? Could be another sign for perceptual decoupling or for sequential processing (see major point 1).

- section 3.1.3. How did you get the BF for the 3-level factor workload? The data analysis section above only says you calculated BF for “pairwise comparisons”.

- section 3.2.1 Again binomial/logit model would have been more consistent and statistically appropriate. Just a suggestion.

- Table S4 and others contain criterion p < .05, different from the main text.

- section 3.4.2 It would have been more consistent to run a mixed model on landing position with the mentioned conditions and saccade latency as continuous predictor. Again, just a suggestion.

- section 3.3.3 Were saccades and blinks entered as fixed effects? If yes, how were they discretised? I recommend considering them either as binary factors (no saccade/blink, at least one saccade/blink), or multi-level factors with polynomial contrast, or numeric predictors. Perhaps the authors want to consider also entering the time interval between last internal-task-related eye movement event saccade task onset as numeric predictor. Here, one would expect a negative effect: larger saccade latency if the eye movement had just been shorter before the saccade target onset, right? But I don’t think the latter is necessary.

- p. 26 “are not the consequence of eye behavior” Instead better “are not _only_ the …”?

- p. 29 “an increase in overall arousal” What about cognitive effort, which is known to be indexed by pupil diameter? Should be discussed.

- Line numbers would have be great!

Christoph Huber-Huber

6. PLOS authors have the option to publish the peer review history of their article (what does this mean?). If published, this will include your full peer review and any attached files.

Reviewer #1: No

Reviewer #2: No

Reviewer #3: **Yes: **Christoph Huber-Huber

---

## [Author Response · Author response to Decision Letter 0]

12 Jun 2023

Below I copied the reviewers comments and my response to them:

Reviewer #1: 

This is a nice, preregistered, eye-movement study exploring the interplay between mental workload and saccades. Participants were asked to complete a mental task (arithmetic or visuo-spatial) with different degrees of load (zero, low, high) and then to perform a saccade towards a target while ignoring a distractor. To sum up, the main results show an influence of the mental task on eye movements, and in particular this was moderate by temporal parameters (SOA), and the type of workload.

I think this paper is clear and well organised. I only have some comments in the hope that these could help the authors.

My main comment is about statistics. It is relatively common to see papers reporting both frequentist and Bayesian analyses, such as in this case. As you performed power analyses, the main results are those concerning the frequentist framework; bayes factors are surely a nice addition, but I am feeling that some readers could be confused; for example, on page 16, you reported that you considered as ‘significant’ a result with p < .001 (I think here you mean .01?) and BF10 > 3. Although the concept of “significance” sounds good for p values, I believe it could be misleading for BF (they actually provide evidence for H0 or H1, depending on the direction). You could think to reframe this part a bit, and the following passages where you refer to statistical ‘significance’.

Response: Thank you for pointing out that the term “significant” is misleading here as it is more commonly associated with the p-value. We reframed the sentence to “We interpret effects that have a p-value below.01 (significant) and a BF10 larger than 3 (weight of evidence in favor of hypothesis).” We also mention this now in the figure notes. Since the term “significant” is more common, in the result section we say that an effect is significant if both p < .01 and BF10 > 3, and if a result has p < .01 but BF10 < 3, we mention that explicitly. 

Another comment is related to methods: On page 13, you say that participants completed another experiment with a different purpose; I can see the practical reasons for that, but I would like to know if all participants completed first the task reported here and then the other task; if not, could tasks order may have played a role?

Response: Half of participants performed the other paradigm before and half after the current paradigm. We now separately check if paradigm order affected pupil diameter, saccade latency, saccade to the target, or maintained fixation. We found no overall effects on eye behavior. Further, we now included paradigm order as fixed effects in all models. In all models, paradigm order had no significant effect and was therefore excluded from the final model. In specific, those were the effects of paradigm order in the models of the following outcome variables: Tonic pupil diameter: t = -1.65, p = .10; pupil diameter change: t = 0.25, p = .81; maintained fixation: z = 0.20, p = .84; first saccade to target: z = 1.18, p = .24; saccade latency: t = -0.23, p = .82.

Other minor comments:

Page 3, paragraph 1.1: I think that, in addition to the citations (11 and 12), it would be fair and relevant to cite two additional papers showing a link between working memory load and microsaccades (Dalmaso et al., 2017).

Page 9: you used a relatively wide range of SOAs; I would appreciate to see some more words explaining this choice and if this was based on previous studies.

References

Dalmaso, M., Castelli, L., Scatturin, P., & Galfano, G. (2017). Working memory load modulates microsaccadic rate. Journal of Vision, 17(3), 6. https://doi.org/10.1167/17.3.6

Response: We added Dalmasio et al., 2017 to the references. 

We now also mention in the methods regarding choice of SOAs: “This range of SOAs was chosen to cover phases of intense internal focus during the task performance (directly after audio around 0.5 and 1s) but also the phase when the internal manipulation is already over (around 2 – 2.5s).” In internal, not published pilots, we determined how long people need for the tasks in order determine time intervals between operations and to make sure SOAs cover the whole task performance (doing the mental navigation/ arithmetic, remembering the result). 

Reviewer #2: 

The authors' study centers around perceptual decoupling during internally directed tasks, with a particular focus on measuring the effects of voluntary saccades on task type and workload. This work builds upon and replicates earlier research by the same authors, with the goal of examining whether voluntary saccade behavior can serve as an indicator of internally directed cognition through the mechanism of perceptual decoupling.

The methods utilized in this study appear to be sound, and the research itself seems to be conducted with rigor. However, there are a few questions that need to be addressed before publication can be granted.

1)

It would be beneficial for the authors to highlight the potential novelty of their study in the introduction, going beyond the goal of simple replication. One way to accomplish this could be to explore whether there are specific reasons why perceptual decoupling with voluntary saccades might manifest differently than in previous reports.

For instance, it may be worth considering the ballistic nature of saccades, which distinguishes them from other types of movements such as manual pointing and reaching. This qualitative difference could potentially lead to distinct patterns in perceptual decoupling, and could further support the originality of the study. By emphasizing these potential novel aspects of their research, the authors can more effectively justify their work and add to the existing body of knowledge in the field.

Response: We now further highlight the novelty of our study in the introduction. E.g., we added the following paragraph:

“

In our previous study, participants continuously performed smooth pursuit eye movements during the internal task. In the current study, the saccade task required participants to maintain fixation on the screen center, perceive the target and distractor, decide which one is the target, plan and execute a saccade there. The nature of eye movement (ballistic vs. continuous) and underlying mechanisms differ between the current and previous study. Hence, the current study tests whether effects of internal task performance found in the previous study also appear for voluntary saccades or show a distinct pattern, and investigate in more detail how the internal task interferes with perception, eye movement preparation and execution.

”

2)

The authors utilized a within-participant experimental design where each participant was involved in all experimental conditions. In the control condition, participants were presented with identical numeric or visuo-spatial input, but without a clear starting point. Specifically, participants received an XX character in the numeric condition and an empty matrix in the visuo-spatial condition.

However, it is important to note that due to transfer effects from the main tasks, participants may have attempted to perform internal operations based on the numeric and visuo-spatial input, even though they were not explicitly instructed to do so. If this were the case, it is possible that a partial and uncontrolled form of perceptual decoupling could have occurred during the control condition as well.

It is important for the authors to consider this potential scenario and its implications for their study. They may want to discuss it in their results section and explore potential ways to control for this confounding factor in future research.

Response: We now address this in the discussion: 

“

In the current study, participants heard the same audio operations in the control condition as in the experimental conditions in order to control for effects of just hearing the audio on saccade latency. Simply hearing the audio could already altered the saccade latency in the saccade task, hence we have no pure measure of how long the saccade latency would be in the saccade task when there is no other paradigm at the same time. In the control condition, participants did receive no starting number (XX instead) and only an empty matrix, and we explicitly asked them to not perform any internal task. It is still possible that they tried to perform the internal task, e.g., out of boredom. However, we see no increase in saccade latency for the first SOAs in the control condition and also task-evoked pupil diameter in the control condition did not show the same pattern as in the low and high load condition. Hence, the data suggests that participants followed instructions and did not perform any internal task during the control condition. Nonetheless, further studies should also involve a control condition without the audios to get an estimate of the saccade latency in a complete single task condition.

”

3)

The quality of the figures appears to be inadequate. It is recommended to generate figures at a higher resolution to enhance their clarity.

Response: Thanks for pointing that out to us. Submission to PLOS ONE requires that we provide figures as separate files in high resolution and the editing software only includes low resolution previews of those figures in the manuscript itself. This might have led to the blurry appearance of the figures in the manuscript. Next to the preview in the manuscript should be a link to the file of the figure. Please check out if the resolution is proper in the figure file.

Reviewer #3: 

It was a pleasure to read this manuscript. The authors present excellent and comprehensive work on how internal tasks (mental arithmetic and spatial planning) interfere with externally directed gaze behaviour. The manuscript is clearly written and showcases outstanding scientific quality in adopting open-science standards and adhering to highly commended power calculation. I just have a few points that probably deserves being discussed in more detail and that might require additional analyses.

Major points

- The main conclusion seems to be that perceptual decoupling delays the saccades in the external task. Although I agree that the pattern or results is consistent with this explanation, I believe that drawing this conclusion is perhaps a bit too pre-mature. The authors show nicely that eye movements associated with internal activities contribute to the delayed saccades effect, but cannot explain it completely. (There is one minor comment about this model, see below.) Thus, there is some residual delay in the saccade latency which, as the authors infer, must be due to perceptual decoupling. Couldn’t it also be that participants, in each trial, conducted first the internal task operation and once this process was finished, the saccade was executed. With a higher load, the internal task operation took more time and therefore, there was a longer delay until the saccade. This explanation would suggest that internal and external tasks were more or less executed one after the other; whereas the perceptual decoupling explanation rather suggests that the saccade target was not perceived that well because participants were internally doing something else, which is a bit different. Even if there are reasons against this sequential tasks explanation, I recommend to discuss it.

Response: 

We now included the following paragraph in the discussion addressing the possibility of a sequential performance of tasks:

“In our previous study [21], participants had to perform smooth pursuit throughout the trial. In the current study, the saccade task required participants to make a decision which object is the target and then execute the saccade there. Hence, the saccade task was more demanding and required a switch of attention from the internal task to the external task. Switching attention between internal and external focus is costly [61]. Could it be that participants first completed the internal task and then performed the saccade task? To achieve the saccade latencies in the current study under the assumption that participants first finished the internal tasks, they would have to finish the internal task in only 0.55 to 0.6 s (target onset at 0.5s + latency – latency of control condition) after operation cue onset, hence already before the end of the audio operation cue. To avoid artefacts of manual responses, we did not assess how long participants needed to perform the internal operations in the present study. However, we had ran pilots in our lab during the planning phase of the paradigms, in which we had assessed response times in a self-paced version of some variations of the internal tasks (without an additional external task). Our pilot participants needed around 1.9 s for the arithmetic task in the high load condition (median = 1.896, SE = 0.143, N = 20), and around 0.8 s for the visuospatial task (low load: median = 0.775, SE = 0.069, N = 12; high load: median = 0.850, SE = 0.072, N = 21). Hence, it seems unlikely that participants first completed the internal task before performing the saccade task. This interpretation is further supported by the fact that saccade latencies still increased after the first SOA. 

The available evidence rather supports the idea that participants interrupted the internal task, switched to the external task and then resumed the internal task. Intense focus on the internal task might have delayed perception of the saccade target and the switch to the external task itself caused delays since switching draws resources also required for the internal tasks [61].

”

[61] Verschooren S, Schindler S, De Raedt R, Pourtois G. Switching attention from internal to external information processing: A review of the literature and empirical support of the resource sharing account. Psychon Bull Rev. 2019;26: 468–490. doi:10.3758/s13423-019-01568-y

- The pupillary response measured with a video-based eye-tracker is per definition affected by the gaze position on screen. The current experimental design balanced the saccade target locations, it is thus expected that any such influence cancels more or less out. However, since quite some trials were excluded, saccade target locations are not balanced anymore in the dataset used in the analysis. Did the authors consider that? see e.g. Gagl et al. (2011) in Behav. Res. Methods for how to deal with this.

Response: Thank you for pointing out how important it is to consider gaze position in pupil data analyses. As mentioned in the Data Preprocessing section, we only used pupil data before saccade target onset, when participants were required to maintain fixation. We now refer to Gagl et al. (2011) regarding the effect of gaze position on pupil data in this paragraph: 

“For analyses of tonic pupil diameter as a measure of overall arousal [47], median pupil diameter in the first 500ms after operation cue onset was used, a phase in which participants were required to maintain fixation. For analyses of phasic pupil diameter (task-evoked pupil response TEPR) as a measure of within-task changes in attentional effort [47], we binned pupil data from operation cue onset until the onset of the saccade stimuli into 500ms time windows and baseline corrected them by subtracting pupil diameter in the 500ms before operation onset [48]. Presentation of the saccade target itself and especially the gaze position change when making a saccade to the target causes changes in pupil diameter [49] and therefore pupil data after the onset of the saccade target were excluded from pupil diameter analyses.”

- General remark on the mixed models: I have the impression that R’s default contrasts were used. They make interaction effects hard to interpret. I recommend to use effect coding or successive difference contrasts, because they make interaction effects interpretable as with a default Anova. I’m not sure, though, whether the anova table with type III SS takes care of that. If it does, no worries.

Response: Thanks for paying close attention to our analyses. We agree that interactions with factors that have more than two levels are hard to interpret from one value alone (in the ANOVA table). As contrasts in the overall model, we used the following approach: for workload, the control condition was the “baseline” to which the low and high condition was compared. For time bin, the first time bin was used as “baseline” to which all the other time bins were compared to. This approach appeared to us to be the most straightforward to understand and required the least manipulation of the model from our side. To follow up interactions, we conducted posttests in which we then systematically investigated each contrast of interest (e.g., low vs. high in the arithmetic task at each time bin). 

- I recommend to report also the models’ random effects structures together with the fixed effects output, not only the Anova tables, perhaps in the supplement.

Response: We previously had the complete model output in the OSF folder. Now, we added tables to the supplemental materials.

- (Just a suggestion because perhaps of interest, if not already known, to the authors.) In the absence of visual input, gaze behaviour seems to reflect a mental number line for numerical tasks. Eyes deviate towards left for smaller and towards right for larger numbers (e.g. Loetscher et al., 2010, Curr. Biol). The authors might want to see whether the addition and subtraction tasks show a similar effect in the the internal-task-related eye movements before saccade target onset.

Response: We added the following to the results section on pre-target saccade activity: 

“

Please note that the overall number of saccades before the saccade task onset was generally low, hence detailed analyses of direction of those saccades was not possible due to low power (e.g., arithmetic task under high load: only 153 trials, visuospatial task under high load : 226 trials per direction from all participants combined). Nonetheless, for interested readers, we included a plot of saccade directions in the supplemental material S1 Fig. S1 Fig does not suggest any systematic covariation of saccade directions relative to task cues (e.g., addition vs subtraction) of those pre-saccade task saccades. Hence, it is possible that typical spontaneous eye movements following the internal number stream or movement through the matrix, respectively, were suppressed when preparing to perform the saccade task.

”

Minor points

- p. 7 typo “; 9)?”

Response: Corrected (collided with the citation format)

- p. 13 Was there really no re-calibration at any point within a task block?

Response: At the beginning of each task block (i.e., after 9-11 trials), hence about every 20-30s, a drift check was performed. If the drift check failed, a recalibration was performed. We now mention that in the methods section. 

- section 2.6 Why a 2x2 ANOVA with workload as 3-level factor? In addition, this line sounds like the analysis here was on proportion correct/error responses. It would have been more consistent to run a mixed model on single trial data not only for saccade latency and pupil but also for task performance (i.e. a binomial model with logit link or similar).

Response: You are right, it is more consistent to use models on single trial data everywhere. We now changed analyses of maintained fixation and whether the saccade landed on the target, to binomial GLMMs. Performance was assessed per block (after 9-11 trials). We tried to calculate binomial GLMMs for performance (correct response) using the block data. However, we were not able to get to a model that converged and had no issues with singularity and even when the model converged it was not possible to obtain any confidence intervals. This was most likely because of the high performance overall, so many participants had all blocks correct and therefore had no variance in their data. So, for analyses of performance (correct response), we returned back to a simple 2 (arithmetic, visuospatial) x 2 (low, high) ANOVA.

- section 2.6. paragraph 2: What was the random intercept for “trial” based on? Was it on something like a trial number, i.e. somehow corresponding to time within the experiment, or was it based on the stimulus item of a trial, thus rather a trial ID, e.g. all trials with saccade target at location 12 o’clock coded with the same trial ID. In any case, trial ID would be the option that makes sense.

Response: We used trial ID as trial intercept to control for effects unique to that trial.

- p. 16 ““significant” if the p-value is below 0.001 and the BF10’s is larger than 3.” This statement is surprising and makes p-values and BF conclusions rather sound incompatible. A p-value of 0.001 is usually associated more with a BF > 20 or similar. I recommend to either stick to p 0.001 and BF 20 or 0.01 and BF 3, or something like that. I see later, e.g. Fig. 2 uses p < .01. Perhaps the earlier .001 was a typo.

Response: Thanks for pointing out that this was confusing. We now state it more clearly (see response to Reviewer 1).

- Fig 2D arithmetic: looks like pupil response started later in the high compared to low and control conditions. Any post-hoc idea why? Could be another sign for perceptual decoupling or for sequential processing (see major point 1).

Response: You are right, in the arithmetic task (but not in the visuospatial task), the increase in pupil diameter from time bin 0-0.5 to 0.5-1s was slightly less steep in the high compared to the low load condition (estimate = -0.02, t = -3.25, p = .014, BF10 = 2,900.73, d = 0.1). The high load condition caught up with the low load condition in the next time bin (1-1.5s). In your earlier point, I already discussed sequential processing. We now added the following to the discussion:

“

In the arithmetic task, the increase in task-evoked pupil diameter from time bin 0-0.5 to 0.5-1s was slightly less steep in the high compared to the low load condition, although the high load condition then caught up and exceeded the low condition. In the arithmetic task, participants had to switch from addition to subtraction for the high load condition, maybe this caused the delay in pupil diameter increase.

”

- section 3.1.3. How did you get the BF for the 3-level factor workload? The data analysis section above only says you calculated BF for “pairwise comparisons”.

Response: Thanks for noting that this information got dropped somewhere during writing/editing. The complete sentence is: “BFs were computed with the BayesFactor package [52] under a default Cauchy prior for each model and pairwise comparison based on the data aggregated across trials.” 

- section 3.2.1 Again binomial/logit model would have been more consistent and statistically appropriate. Just a suggestion.

Response: We now use binomial model for analysis of maintained fixation.

- Table S4 and others contain criterion p < .05, different from the main text.

Response: Thanks for noticing the typos. It shows how thoroughly and devoted you read our manuscript in order to provide us with valuable suggestions. We corrected the typos (now p <.01)

- section 3.4.2 It would have been more consistent to run a mixed model on landing position with the mentioned conditions and saccade latency as continuous predictor. Again, just a suggestion.

Response: We now use binomial mixed models for correct saccade to target. See above.

- section 3.3.3 Were saccades and blinks entered as fixed effects? If yes, how were they discretised? I recommend considering them either as binary factors (no saccade/blink, at least one saccade/blink), or multi-level factors with polynomial contrast, or numeric predictors. Perhaps the authors want to consider also entering the time interval between last internal-task-related eye movement event saccade task onset as numeric predictor. Here, one would expect a negative effect: larger saccade latency if the eye movement had just been shorter before the saccade target onset, right? But I don’t think the latter is necessary.

Response: Saccades and blinks were first dichotomized (there were so few trials with more than one blink or saccade, hence we used 0 = no saccade/blink and 1 = any number of saccades/blinks) and entered as fixed effects. Since the majority of trials still had no blink or saccade before the saccade target appearance, we refrained from calculating the latter analyses due to low power. But as you suggested, we would also expect that the saccade latency would be longer the closer a previous saccade or blink was.

- p. 26 “are not the consequence of eye behavior” Instead better “are not _only_ the …”?

Response: Done.

- p. 29 “an increase in overall arousal” What about cognitive effort, which is known to be indexed by pupil diameter? Should be discussed.

Response: We adapted the sentence accordingly.

- Line numbers would have be great!

Response: added

---

## [Decision Letter · Decision Letter 1]

21 Jul 2023

PONE-D-23-07956R1The effects of type and workload of internal tasks on voluntary saccades in a target-distractor saccade taskPLOS ONE

Dear Dr. Walcher,

Thank you for submitting your manuscript to PLOS ONE. After careful consideration, we feel that it has merit but does not fully meet PLOS ONE’s publication criteria as it currently stands. Therefore, we invite you to submit a revised version of the manuscript that addresses the points raised during the review process.

We look forward to receiving your revised manuscript.

Kind regards,

Jie Wang, Ph.D.

Academic Editor

PLOS ONE

Journal Requirements:

Reviewers' comments:

Reviewer's Responses to Questions

**Comments to the Author**

1. If the authors have adequately addressed your comments raised in a previous round of review and you feel that this manuscript is now acceptable for publication, you may indicate that here to bypass the “Comments to the Author” section, enter your conflict of interest statement in the “Confidential to Editor” section, and submit your "Accept" recommendation.

Reviewer #1: All comments have been addressed

Reviewer #2: All comments have been addressed

Reviewer #3: (No Response)

2. Is the manuscript technically sound, and do the data support the conclusions?

Reviewer #1: Yes

Reviewer #2: Yes

Reviewer #3: Yes

3. Has the statistical analysis been performed appropriately and rigorously? 

Reviewer #1: Yes

Reviewer #2: Yes

Reviewer #3: Yes

4. Have the authors made all data underlying the findings in their manuscript fully available?

Reviewer #1: Yes

Reviewer #2: Yes

Reviewer #3: Yes

5. Is the manuscript presented in an intelligible fashion and written in standard English?

Reviewer #1: Yes

Reviewer #2: Yes

Reviewer #3: Yes

6. Review Comments to the Author

Reviewer #1: I am happy with the revised version of this manuscript. I can therefore suggest its publication. I thank the authors for having considered all my comments.

Reviewer #2: All my comments have been addressed. I have no further comments for the authors.

The manuscript is technically sound, and the data support the conclusions.

Reviewer #3: Many thanks for carefully addressing all my concerns. Eventually, I only have a very minor request related to the binomial GLMM for task performance. Here, the grouping structure of the data with the blocks is unclear to me. It sounds like you calculated a summary statistic (like mean performance across a block) and this statistic was your base level for the statistical model, but this approach would be incompatible with a binomial model. So, I guess you did indeed take individual trial data. It surprises me that it was that hard to get the model to converge. In such a case and with that many correct trials, I would consequently also rather not draw too strong conclusions from the ANOVA results and recommend to add a short paragraph to the results section that describes your logic for why you chose the ANOVA.

7. PLOS authors have the option to publish the peer review history of their article (what does this mean?). If published, this will include your full peer review and any attached files.

Reviewer #1: No

Reviewer #2: No

Reviewer #3: **Yes: **Christoph Huber-Huber

---

## [Author Response · Author response to Decision Letter 1]

4 Aug 2023

Reviewer #3: Many thanks for carefully addressing all my concerns. Eventually, I only have a very minor request related to the binomial GLMM for task performance. Here, the grouping structure of the data with the blocks is unclear to me. It sounds like you calculated a summary statistic (like mean performance across a block) and this statistic was your base level for the statistical model, but this approach would be incompatible with a binomial model. So, I guess you did indeed take individual trial data. It surprises me that it was that hard to get the model to converge. In such a case and with that many correct trials, I would consequently also rather not draw too strong conclusions from the ANOVA results and recommend to add a short paragraph to the results section that describes your logic for why you chose the ANOVA.

Response: Thank you for pointing out, that it was not clear in the manuscript that the performance was measured at the end of a block, hence after 9-11 trials of audio directions/numbers (see Figure 1) and not after each trial. We therefore used this binomial measure of performance per block (not trial) as input for the binomial GLMM of performance. We now extended the note below Figure 1 with the info in the brackets: “At the end of each block, participants were asked to report the result of the internal task (this report is used as the performance measure).” And also in the methods section we included brackets, e.g., in line 238: “At the end of each arithmetic task block (after 9-11 trials), participants had to type in the result.”

Further, in the analysis strategy and results section, we also mention more clearly why there is a measure per block and not per trial for performance analysis:

Note of Figure 2, added brackets: “The proportion of blocks with correct responses in the internal tasks (participants had to report the final result after 9-11 trials at the end of a block, see Fig 1).”

In the methods section, at “Analysis strategy”,:

“For internal task performance, we calculated binomial GLMMs, using the response per block (correct, incorrect). Participants reported the result of the arithmetic task, and the final position of the patch in the matrix, respectively, at the end of a block, hence after 9-11 trials, leading to one performance measure per block, not trial, see Fig 1.”

And we now include in the results section what we previously only included in our response to your comments in the last round of revision:

“We tried to calculate binomial GLMMs for internal task performance, using the response per block (correct, incorrect). (Participants reported the result of the arithmetic task, and the final position of the patch in the matrix, respectively, at the end of a block, hence after 9-11 trials, leading to one performance measure per block, not trial, see Fig 1.). However, the models did not converge and had singularity. We therefore reverted to a repeated-measures ANOVA using the proportion of correct blocks per Task and Workload.”

Since we had block data instead of trial data for the binomial GLMM of performance, the overall number of observations is much lower for this analysis, potentially explaining why these GLMMs did not converge.

---

## [Decision Letter · Decision Letter 2]

7 Aug 2023

The effects of type and workload of internal tasks on voluntary saccades in a target-distractor saccade task

PONE-D-23-07956R2

Dear Dr. Walcher,

We’re pleased to inform you that your manuscript has been judged scientifically suitable for publication and will be formally accepted for publication once it meets all outstanding technical requirements.

Kind regards,

Jie Wang, Ph.D.

Academic Editor

PLOS ONE

Additional Editor Comments (optional):

Reviewers' comments:

Reviewer's Responses to Questions

**Comments to the Author**

1. If the authors have adequately addressed your comments raised in a previous round of review and you feel that this manuscript is now acceptable for publication, you may indicate that here to bypass the “Comments to the Author” section, enter your conflict of interest statement in the “Confidential to Editor” section, and submit your "Accept" recommendation.

Reviewer #1: All comments have been addressed

Reviewer #3: All comments have been addressed

2. Is the manuscript technically sound, and do the data support the conclusions?

Reviewer #1: Yes

Reviewer #3: Yes

3. Has the statistical analysis been performed appropriately and rigorously? 

Reviewer #1: Yes

Reviewer #3: Yes

4. Have the authors made all data underlying the findings in their manuscript fully available?

Reviewer #1: Yes

Reviewer #3: Yes

5. Is the manuscript presented in an intelligible fashion and written in standard English?

Reviewer #1: Yes

Reviewer #3: Yes

6. Review Comments to the Author

Reviewer #1: I am Happy with this version and I suggest the publication of this paper in this journal. All my comments have been addressed.

Reviewer #3: (No Response)

7. PLOS authors have the option to publish the peer review history of their article (what does this mean?). If published, this will include your full peer review and any attached files.

Reviewer #1: No

Reviewer #3: **Yes: **Christoph Huber-Huber

---

## [Editor Report · Acceptance letter]

15 Aug 2023

PONE-D-23-07956R2 

The effects of type and workload of internal tasks on voluntary saccades in a target-distractor saccade task 

Dear Dr. Walcher:

I'm pleased to inform you that your manuscript has been deemed suitable for publication in PLOS ONE. Congratulations! Your manuscript is now with our production department. 

Kind regards, 

on behalf of

Dr. Jie Wang 

Academic Editor

PLOS ONE